# Brick wall quantum circuits with global fermionic symmetry

Pietro Richelli[1,2,3], Kareljan Schoutens[1,2] and Alberto Zorzato[1,2*]

**1** Institute for Theoretical Physics, University of Amsterdam,
Science Park 904, 1098 XH Amsterdam, The Netherlands
**2** QuSoft, Science Park 123, 1098 XG Amsterdam, The Netherlands
**3** Kavli Institute of Nanoscience, Delft University of Technology,
Lorentzweg 1, 2628 CJ, Delft, the Netherlands

⋆ a.zorzato@uva.nl

## Abstract

We study brick wall quantum circuits enjoying a global fermionic symmetry. The constituent 2-qubit gate, and its fermionic symmetry, derive from a 2-particle scattering matrix in integrable, supersymmetric quantum field theory in 1+1 dimensions. Our 2-qubit gate, as a function of three free parameters, is of so-called free fermionic or matchgate form, allowing us to derive the spectral structure of both the brick wall unitary $U_F$ and its, non-trivial, Hamiltonian limit $H_\gamma$ in closed form. We find that the fermionic symmetry pins $H_\gamma$ to a surface of critical points, whereas breaking that symmetry leads to non-trivial topological phases. We briefly explore quench dynamics for this class of circuits.

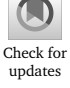

# 1 Introduction

## 1.1 Brick wall circuits with global fermionic symmetry

A particularly fascinating interface of quantum information and quantum many-body physics is the study of quantum circuits that represent the (unitary) time evolution of systems in quantum particle or material physics. In their most basic form these circuits take the form of a 'brick wall' circuit whose properties are set by the choice of a 2-qubit gate that represents a single brick in the wall. Studies of this type have typically opted for one of two extreme choices: either one assumes randomly chosen 2-qubit unitaries ([1] and references therein), or, on the opposite, one picks a structured 2-qubit gate that leads to a degree of analytical control of the unitary brick wall (UBW) circuit.

Indeed, if the 2-qubit gate is chosen to be a so-called $R$-matrix satisfying a Yang-Baxter identity, one can arrange a corresponding UBW circuit such that, as an operator, it commutes with a large number of conserved charges. See [2–4] where this procedure was proposed and analyzed and [5–7] where such circuits, and an array of physical phenomena related to 'integrable trotterization', were studied. Ref. [8] in particular has implemented these ideas for the $R$-matrix of the XXX integrable spin-1/2 Heisenberg magnet and analyzed its conserved charges, both analytically and in realizations on quantum computing hardware. We point out other experiments exploiting similar concepts [9, 10].

In this paper we make a different choice for a highly structured brick in the wall: we will analyze UBW circuits built from 2-qubit gates with a free fermionic structure, known as 'matchgates' in the quantum information literature. This structure gives a high degree of analytical control, allowing us to make precise statements on the spectral structures associated with our circuits, both for the brick wall unitary $\mathbf{U_F}$ and for a class of Hamiltonians $\mathbf{H}_\gamma$ that we associate with these circuits. At the same time, it leads a rich structure for the quantum dynamics of $\mathbf{U_F}$ and the phase diagram of $\mathbf{H}_\gamma$.

The construction and analysis of these fermionic UBW circuits poses various challenges and open questions, which we address in this paper. First, a consistent interpretation of the qubit states $|0\rangle$ and $|1\rangle$ as fermionic states $|b\rangle$ and $|f\rangle$ requires that we equip the multi-qubit Hilbert space with a graded tensor product. Having made this interpretation, it is natural to consider a fermionic symmetry that rotates $|b\rangle$ into $|f\rangle$ and that commutes with the brick wall unitary $\mathbf{U_F}$. We will denote such a symmetry as a global fermionic symmetry. In section 1.2 below we use a connection with supersymmetric particle scattering theories in 1+1 dimensions to identify a 3-parameter class $\check{\mathbf{S}}(\alpha, \gamma, \theta)$ of 2-qubit gates which are of free fermionic form and satisfy a global fermionic symmetry. In this connection, $\theta$ corresponds to the difference in rapidity of the particles scattering in 1+1D, $\gamma$ corresponds to the logarithm of the ratio of the masses $m_1$ and $m_2$ of the particles on even and odd lines, and $\alpha$ corresponds to the strength of the interactions in the 1+1D scattering theory.

A second challenge is to understand the implications of the free fermionic structure and the global fermionic symmetry for the Hamiltonians $\mathbf{H}_\gamma$ associated to our circuits. Interestingly, these Hamiltonians take the form of staggered versions of the Kitaev chain model, describing the pairing of spinless fermions on a 1D lattice. We will find that the global fermionic symmetry precisely pitches these Hamiltonians to critical points, separating a variety of topological phases in the BDI class.

A third challenge is to understand (quench) dynamics set by $\mathbf{U_F}(\alpha, \gamma, \theta)$. In this, a particular role is played by the mass ratio $m_1/m_2$ : we find that, in a large part of parameter space, the UBW dynamics has a characteristic drift velocity

$$v_d = 2\frac{m_1 - m_2}{m_1 + m_2},\tag{1}$$

on a scale where $v = 2$ is the maximal velocity set by the circuit geometry (2 steps left or right over one odd and one even layer of the circuit).

It is interesting to compare our findings for the case of unequal masses $(m_1, m_2)$ to the results in a recent study of Zadnik et al. [11], which analyzes a family of quantum UBW circuits constructed out of $SU(2)$-symmetric unitary gates In these circuits, the odd and even qubit lines carry $SU(2)$ representations $(s_1, s_2)$. The authors point out non-trivial transport properties of this system due to the presence of individually broken time-reversal and space-reflection symmetries, but a combined $\mathcal{PT}$ symmetry. These properties include a drift velocity with value

$$v_d = \frac{C_1 - C_2}{C_1 + C_2},\tag{2}$$

with $C_i = s_i(s_i + 1)$ the Casimir of the corresponding $SU(2)$ representation.

Further motivation for our study is given by a growing interest in the study of Quantum Cellular Automata (QCAs), our finite depth quantum circuit being an example. QCAs are quantum generalisations of cellular automata, and they can be interpreted as regularizations of continuous quantum field theories on discrete spacetime obeying strict causality and unitarity (see [12, 13] and references therein). We build on this connection by studying the consequences of global fermionic constraints on a supersymmetric quantum field theory regularized on a lattice.

## 1.2 2-qubit gates from supersymmetric particle scattering

A context where solutions to a Yang-Baxter equation arise in a natural way is that of factorizable particle scattering in massive integrable quantum field theories (QFT) in 1+1 dimensions. These theories describe (massive) particles in 1+1D, whose many-body scattering processes factor into 2-body processes as a consequence of integrability. In the early 1990's a great number of non-trivial such $S$-matrices were identified through the study of QFT's arising from relevant, integrable perturbations of conformal field theories (CFT), or from direct analysis of integrable QFT's such as the sine-Gordon model. A highlight of the former approach has been the identification of the $S$-matrices for massive particles arising from the magnetic perturbation of the Ising CFT, with the masses and the scattering matrices all organized by an underlying $E_8$ symmetry [14].

In 1+1D QFT context the notion of a fermionic symmetry of a 2-body scattering matrix takes the form of space-time supersymmetry [15]. It expresses the 1+1D super Poincaré algebra, in the form

$$\mathcal{Q}^2 = \mathcal{P}, \qquad \overline{\mathcal{Q}}^2 = \overline{\mathcal{P}}, \qquad \mathcal{P}^0 = \mathcal{P} + \overline{\mathcal{P}}, \qquad \mathcal{P}^1 = \mathcal{P} - \bar{\mathcal{P}}, \tag{3}$$

with the energy $\mathcal{P}^0$ and momentum $\mathcal{P}^1$ operators taking values

$$p^0 = m \cosh(\theta), \qquad p^1 = m \sinh(\theta), \tag{4}$$

for on-shell particles with mass $m_i$ and rapidity $\theta_i$. In such supersymmetric QFT's, particles come in doublets $(b_i, f_i)$ of mass $m_i$, giving rise to (asymptotic) particle states

$$|A_{i_1}(\theta_1) \ldots A_{i_n}(\theta_n)\rangle, \tag{5}$$

with $A_i = b_i, f_i$. A particularly natural choice for the representation of the supercharges satisfying the algebra of eq. 3 is, in matrix form,

$$Q = \sigma_x, \qquad \overline{Q} = \sigma_y, \qquad Q_L = \sigma_z, \tag{6}$$

with $Q_L$ representing the parity operator. On a 2-particle state the supercharges then act as

$$\mathbf{Q^l} = \sqrt{m_1}\, e^{\theta_1/2} Q \otimes \mathrm{I} + \sqrt{m_2}\, e^{\theta_2/2} Q_L \otimes Q, \tag{7}$$

and similar for $\mathbf{Q^r}$ (in terms of $\overline{Q}$ instead of $Q$ and minus signs in front of $\theta_1$ and $\theta_2$). Note that the operator $Q_L$ in the second term expresses the fermionic nature of the supercharges.

The paper [15] identified the most general 2-body $S$-matrix commuting with this particular representation of space-time supersymmetry. Written on the basis $|b_i(\theta_1)b_j(\theta_2)\rangle$, $|b_i(\theta_1)f_j(\theta_2)\rangle$, $|f_i(\theta_1)b_j(\theta_2)\rangle$, $|f_i(\theta_1)f_j(\theta_2)\rangle$, it takes the form

$$\check{\mathbf{S}}(\theta) = f(\theta) \begin{pmatrix} 1 - t\tilde{t} & 0 & 0 & t + \tilde{t} \\ 0 & 1 + t\tilde{t} & t - \tilde{t} & 0 \\ 0 & -t + \tilde{t} & 1 + t\tilde{t} & 0 \\ -t - \tilde{t} & 0 & 0 & 1 - t\tilde{t} \end{pmatrix} + g(\theta) \begin{pmatrix} 1 & 0 & 0 & 0 \\ 0 & 0 & 1 & 0 \\ 0 & 1 & 0 & 0 \\ 0 & 0 & 0 & -1 \end{pmatrix}. \tag{8}$$

We point out how the left matrix in eq.8 corresponds, in statistical mechanics terms, to the R-matrix of an 8-vertex model in the presence of an external magnetic field [16–18]. The dependence on $\theta = \theta_1 - \theta_2$ and the different masses is through the parameters

$$t = \tanh\left[\frac{\theta + \gamma}{4}\right], \qquad \tilde{t} = \tanh\left[\frac{\theta - \gamma}{4}\right], \quad \text{with} \quad \gamma = \log\left(\frac{m_i}{m_j}\right). \tag{9}$$

The functions $f(\theta)$ and $g(\theta)$ are related as

$$f(\theta) = \frac{\alpha}{2i} \sqrt{m_i m_j} \frac{\cosh(\theta/2) + \cosh(\gamma/2)}{\cosh(\theta/2)\sinh(\theta/2)} g(\theta). \tag{10}$$

The parameter $\alpha > 0$ sets the strength of the Bose-Fermi mixing interactions: for $\alpha = 0$ the transformations $|b\rangle \to |f\rangle$ and $|f\rangle \to |b\rangle$ are not allowed, and the scattering matrix reduces to a graded permutation, $\check{\mathbf{S}} = \mathbf{\Pi}$. The overall normalization $g(\theta)$ provided in [15] is particular to the 1+1D scattering context and not important here. For $\check{\mathbf{S}}(\theta)$ to be unitary we choose $g(\theta)$ as

$$g(\theta) = i \left( 1 + 2\alpha^2 \frac{\cosh(\theta) + \cosh(\gamma)}{\sinh^2(\theta)} \right)^{-1/2}. \tag{11}$$

The paper [15] showed that, without any further restrictions, these scattering matrices satisfy the Yang-Baxter relation. The same paper identified concrete examples of integrable supersymmetric perturbations of supersymmetric CFT, with $S$-matrices given by eq. 8. The simplest, featuring a single doublet $(b, f)$, starts from a CFT with central charge $c = -21/4$, and provides a supersymmetric analogue of the scattering theory arising from a perturbation of the CFT going with the Yang-Lee edge singularity at $c = -22/5$ [19]. A later paper [20] confirmed this identification with the help of a Thermodynamic Bethe Ansatz (TBA) procedure.

Back to the subject of the current paper: taking $\check{\mathbf{S}}(\theta)$ (properly normalized) as our fundamental 2-qubit gate, we build and analyze a class of UBW circuits that are naturally endowed with a graded tensor product structure, integrability and a global fermionic symmetry having its origin in space-time supersymmetry in 1+1D.

Before delving into this, we remark that, if we wish to study UBW circuits with open boundary conditions (OBC), we will need boundary operators that respect the fermionic symmetry. Such boundary reflection operators were found and studied in [21]. On the 1-particle states $|b(\theta)\rangle$, $|f(\theta)\rangle$ they take the concrete form

$$\mathbf{K}(\theta) = \sqrt{\frac{2}{\cosh(\theta)}} \begin{pmatrix} \cosh(\theta/2 - i\pi/4) & 0 \\ 0 & \cosh(\theta/2 + i\pi/4) \end{pmatrix}, \tag{12}$$

and satisfy

$$\tilde{\mathbf{Q}}(-\theta)\mathbf{K}(\theta/2) = \mathbf{K}(\theta/2)\tilde{\mathbf{Q}}(\theta), \tag{13}$$

with $\tilde{\mathbf{Q}} = \tilde{\mathbf{Q}}^{\mathbf{l}} + \tilde{\mathbf{Q}}^{\mathbf{r}}$, where $\tilde{\mathbf{Q}}^{\mathbf{l}} = e^{\theta/2}\sigma_x$, $\tilde{\mathbf{Q}}^{\mathbf{r}} = e^{-\theta/2}\sigma_y$ are the single site representations of the supercharge operator (as explained in [21]). Thus, the boundary reflection operators "commute" with the sum of the left and right supercharges.

## 1.3 Free fermion structure and matchgate condition

It has been observed and employed early on [20, 22–24] that the matrix $\check{\mathbf{S}}(\theta)$ in eq. 8, parameterized by $\alpha$ and $\gamma$ and viewed in the context of a statistical mechanics model, satisfies a 'free fermion condition'. In the language of quantum information, it is said that the corresponding 2-qubit gate is a so-called matchgate. Following the definition in [25], a matchgate is a 2-qubit gate which in the computational basis takes the form

$$G(A,B) = \begin{pmatrix} p & 0 & 0 & q \\ 0 & w & x & 0 \\ 0 & y & z & 0 \\ r & 0 & 0 & s \end{pmatrix}, \qquad A = \begin{pmatrix} p & q \\ r & s \end{pmatrix}, \qquad B = \begin{pmatrix} w & x \\ y & z \end{pmatrix}, \tag{14}$$

where $A$ and $B$ are both elements of $SU(2)$ or $U(2)$ and they have the same determinant. Our matrix $\check{\mathbf{S}}(\theta)$ is precisely of this form.

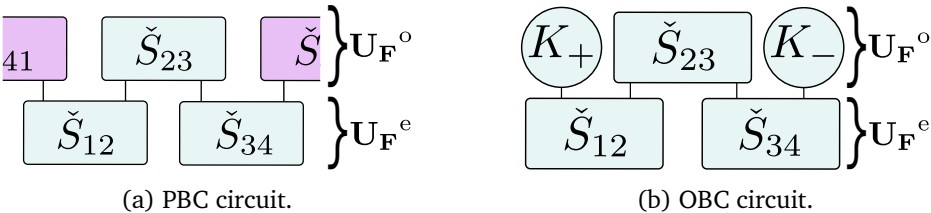

(a) PBC circuit.   (b) OBC circuit.

Figure 1: PBC/OBC single-layer circuits for a 4-site system.

The free fermion or matchgate condition guarantees that the matrix $\check{\mathbf{S}}(\theta)$ can be written as the exponent of a bilinear in free fermion creation and annihilation operators, see eq. 33 below.

This observation explains the fact that $\check{\mathbf{S}}(\theta)$ satisfies the Yang-Baxter relations. It also confirms the original observation that the scattering matrix $\check{\mathbf{S}}(\theta)$ can be understood in fermionic language. Naturally the map between bosonic qubits (spins) and spinless fermions is a Jordan-Wigner transformation. This is a concern in particular in the case where periodic boundary condition (PBC) are imposed, as the matrix $\check{\mathbf{S}}(\theta)_{L1}$ (see figure 1a) connecting the outer qubits numbered as $L$ and 1 depends on the total parity of the state carried by qubits $2, \ldots, L-1$. See section 2.2 for a discussion of this point.

The free fermion or matchgate condition also implies that (under mild conditions, see [25]) UBW circuits based on $\check{\mathbf{S}}(\theta)$ can be simulated efficiently in polynomial time on a classical computer [25–27]. In addition, the unitary operator represented by a UBW can be represented as

$$\mathbf{U_F}(\theta) = \exp[i\mathbf{E}(\theta)] = \exp\left[i\sum_j \epsilon_j \left(\eta_j^\dagger \eta_j - \frac{1}{2}\right)\right], \tag{15}$$

where $\eta_j^\dagger$ and $\eta_j$ are linear combinations of the 1-particle creation and annihilation operators $c_j^\dagger$ and $c_j$, and the $\epsilon_j$ are 1-particle energies. Explicitly analyzing this expression in a momentum basis, and computing the dispersion relations $\epsilon_j^k$, is among the most important goals of this paper.

We remark that the matrices $\check{\mathbf{S}}(\theta)$, which we identified by imposing a global fermionic symmetry, are not the most general matrices satisfying a free fermion or matchgate condition. We point out [28] for a recent analysis of quantum circuits with underlying free fermionic structure. We can thus study the breaking of the global fermionic symmetry, inherited from the SUSY CFT, without leaving the space of free fermion (matchgate) gates. As explained below, we will find that, typically, imposing the global fermionic symmetry will force a gapless dispersion and hence critical behaviour in the corresponding quantum dynamics. Breaking that symmetry typically leads to topologically non-trivial phases protected by a gap.

UBW circuits based on an $R$-matrix of XXZ type similarly have a free fermion point. That point enjoys a $U(1)$ symmetry, allowing to solve for the dispersions $\epsilon_i^k$ by solving a quadratic equation [5]. In our situation only a $\mathbb{Z}_2$ symmetry (the fermion parity) survives, implying that our dispersions obey quartic equations at best. This then translates into a richer dependence of the $\epsilon_i^k$ on the momentum $k$. As an example, we identify situations with two critical modes, one with dispersion linear in $k$ and the other with cubic ($\propto k^3$) dispersion for a specific choice of parameters.

## 1.4 Organization of the paper

Section 2 presents a number of preliminaries needed to set up the fermionic brick wall circuits. We present the graded tensor product structure underlying fermionic quantum circuits, define

the brick wall unitaries $\mathbf{U_F}$ for periodic and open boundary conditions, and specify the action of global fermionic symmetries $\mathbf{Q^L}$ and $\mathbf{Q^R}$. We present an explicit free fermion form of the scattering matrix $\check{\mathbf{S}}(\theta)$ and define the Hamiltonian $\mathbf{H}_\gamma$.

Section 3 is devoted to a detailed analysis in the Hamiltonian limit described by $\mathbf{H}_\gamma$, while section 4 presents the spectral analysis of the unitary $\mathbf{U_F}$ for periodic boundary conditions. For both these cases we manage to express the single-layer dispersions $\epsilon_i^k$ in terms of $\alpha$, $\gamma$, $\theta$ and $k$ in closed-form (section 3) or close-to-closed form (section 4). In section 5 we explore quench dynamics of $\mathbf{U_F}$, starting from the all-0 state or from a state with a single seed '1' in the background of the all-0 state. Using a TEBD algorithm, we track the polarizations $\sigma_i^z$ after applying $N_l$ layers of $\mathbf{U_F}$. We back this up by analytical reasoning based on the free fermionic spectral structure found in section 4, and reproduce the equilibrium values of the polarizations from a free fermion generalized Gibbs ensemble (GGE).

Section 6 has some conclusions and a brief outlook on further research. Appendix A presents a graded extension of the Floquet Baxterization theorem for integrable quantum circuits, appendix B provides some of the details of our analysis of the spectral structure of $\mathbf{H}_\gamma$ in section 3, and appendix C discusses quench dynamics of $\mathbf{U_F}$ with PBC, exploiting the free fermionic structure.

# 2 Brick wall circuits: Preliminaries

Quantum brick wall circuits have been studied in depth in recent years. Here we propose their fermionic extension, meaning that we treat our multi-qubit register as a graded Hilbert space and trade the Pauli spin operators for their fermionic counterparts.

## 2.1 Graded tensor product

In order to properly introduce what we will call a graded brick wall we first need to outline the main properties of graded vector spaces, the content of this subsection is mainly taken from [29]. We will focus on 2 dimensional graded vector spaces $V^g \cong \mathbb{C}^{(1|1)} \cong \mathbb{C} \oplus \mathbb{C}$ with basis $\{e_0, e_1\}$. A graded space is characterized by a parity function $p : V^g \to \mathbb{Z}_2$ that acts on the basis space as:

$$p(e_0) = 0, \qquad p(e_1) = 1. \tag{16}$$

Naturally the permutation operator defined on graded spaces is different than the usual, thus we will denote it with $\mathbf{\Pi}$. On two vectors $v, w \in \mathbb{C}^{(1|1)}$ acts in the following way:

$$\mathbf{\Pi}(e_i \otimes e_j) = (-1)^{p(e_i)p(e_j)} e_j \otimes e_i, \qquad \forall i, j \in \mathbb{Z}_2. \tag{17}$$

The last necessary notion about graded vector spaces is the graded tensor product. The parity operator is essential to define the tensor product in graded spaces, in fact taken two vectors $v, w \in \mathbb{C}^{(1|1)}$ then their tensor product will live in the space $\mathbb{C}^{(1|1)} \otimes \mathbb{C}^{(1|1)}$ and it will be:

$$v \otimes w = v_i e_i \otimes w_j e_j = (e_i \otimes e_j) v_i w_j (-1)^{p(e_i)p(e_j)}. \tag{18}$$

To make the notation lighter from now on we will refer to the basis $\{e_0, e_1\}$ as $\{0, 1\}$. The graded tensor product can be extended on the space of endomorphisms $\mathrm{End}(\mathcal{H})$, with $\mathcal{H} \cong \bigotimes_i V_i^g$. Given $\mathcal{H} \cong V_1^g \otimes \cdots \otimes V_N^g$, a matrix element of a local operator on the subspace $V_i^g$ is defined as:

$$\left(\hat{O}_i\right)_{\mathbf{b}}^{\mathbf{a}} = (-1)^{\sum_{j<i} p(a_j)(a_i+b_i)} \left(\mathbb{1} \otimes \cdots \otimes \mathbb{1} \otimes \hat{O}_i \otimes \mathbb{1} \otimes \cdots \otimes \mathbb{1}\right)_{\mathbf{b}}^{\mathbf{a}}, \tag{19}$$

where $\mathbf{a} = (a_1, \ldots, a_n)$ and $\mathbf{b} = (b_1, \ldots, b_n)$ are, respectively, two vectors in the computational basis $a_i, b_j \in \{0, 1\}$ (e.g. for 4 qubits $\mathbf{a} = (0, 1, 1, 0)$), that denote the row and column elements

of the matrix representation of the operator. Each element of the these two vectors will label the state of the subspace $V_i^g$. This formalism is also used in the definition of fermionic MPS [30]. Using this notation we can extend the parity operator to $\mathrm{End}(\mathcal{H})$ with $\mathcal{H} \cong \bigotimes_i V_i^g$ as:

**Definition 2.1** (Operator parity). Given an operator $S \in \mathrm{End}\left(V_1^g \otimes \cdots \otimes V_n^g\right)$ its parity is defined as:

$$p\left(S_{\mathbf{b}}^{\mathbf{a}}\right) = \sum_{i=1}^{n} p(a_i) + p(b_i) \mod 2. \tag{20}$$

It is important to stress that parity is a property only of the states and operators defined on graded Hilbert spaces, for the purpose of writing down a quantum circuit of graded operators we need to work with their representation into a non graded space. The representation can be easily found by evaluating the graded tensor product of a generic local operator. Any operator on the $\mathbb{Z}_2$ graded space can be written as a linear combination of an even an an odd part $\hat{O} = \alpha \hat{O}^e + \beta \hat{O}^o$. The representation of the even part is simply:

$$\hat{O}_i^e = \mathbb{1} \otimes \cdots \otimes \mathbb{1} \otimes \hat{O}_i^e \otimes \mathbb{1} \otimes \cdots \otimes \mathbb{1}, \tag{21}$$

while for the odd part $p(\hat{O}) = 1$,

$$\hat{O}_i^o = \sigma^z \otimes \cdots \otimes \sigma^z \otimes \hat{O}_i^o \otimes \mathbb{1} \otimes \cdots \otimes \mathbb{1}. \tag{22}$$

Therefore in our brick wall there will be strings of $\sigma^z$ operator that appear in the ungraded representation of the odd graded operators. Although seemingly a complication, the non locality that will characterize our circuit is the same we have to deal with once we take a Jordan-Wigner transformation. The boundary term $(-1)^{\hat{N}}$ that naturally appears in the Jordan-Wigner transformation will be cancelled by the string of $\sigma^z$ given by the graded tensor product. This peculiar feature makes our system fermionic and it shows how the supersymmetry present in the field theory translates into what we call global fermionic symmetry.

## 2.2 PBC: UBW and graded Floquet Baxterization

To set up our brick wall circuits, we assume particles of mass $m_1$ and rapidity $\theta_1 = \theta/2$ on the odd qubit lines, and particles of mass $m_2$ and rapidity $\theta_2 = -\theta/2$ on the even lines, in agreement with [31]. This leads to the following unitary operator, which we will loosely refer to as a time evolution operator,

$$\mathbf{U_F}(\alpha, \gamma, \theta) = \mathbf{U_F}^o(\alpha, \gamma, \theta)\mathbf{U_F}^e(\alpha, \gamma, \theta) = \left(\prod_{i=1}^{L/2} \check{\mathbf{S}}_{2i,2i+1}(\theta)\right)\left(\prod_{i=1}^{L/2} \check{\mathbf{S}}_{2i-1,2i}(\theta)\right), \tag{23}$$

where $\check{\mathbf{S}}(\theta)$ is the unitary gate defined by equation 8, with $\theta = \theta_1 - \theta_2$. We put $m_1 m_2 = 1$, absorbing the overall mass scale in $\alpha$. We will refer to a single application of $\mathbf{U_F}(\alpha, \gamma, \theta)$ as one single layer, comprised of two sublayers (even $\mathbf{U_F}^e(\alpha, \gamma, \theta)$ and odd $\mathbf{U_F}^o(\alpha, \gamma, \theta)$).

In [31] it was shown that a brick wall circuit of this type is integrable if $\mathbf{S} = \mathbf{P}\check{\mathbf{S}}$ satisfies the Yang-Baxter equations. Thus a transfer matrix can be constructed as

$$\mathbf{t}(u, \theta_1, \theta_2) = \mathrm{Tr}_a\left[\prod_i^L \mathbf{S}_{a,i}(u - \theta_j)\right], \quad j = i \mod 2 + 1, \tag{24}$$

such that is satisfies the following commutation relations

$$[\mathbf{t}(u, \theta_1, \theta_2), \mathbf{t}(v, \theta_1, \theta_2)] = 0, \qquad [\mathbf{U_F}(\theta), \mathbf{t}(u, \theta_1, \theta_2)] = 0, \quad \forall u, v \in \mathbb{C}. \tag{25}$$

The integrability of the brick wall circuit was proved only for standard Hilbert spaces without grading, therefore we need to extend this result to graded Hilbert spaces. The statement and proof of a *graded Floquet Baxterization theorem* can be found in Appendix A. Since the scattering matrix 8 is free fermionic we will not need the theorem in the current analysis, but it will be useful for future works on non free-fermionic models.

The scattering matrix is defined on the tensor product of graded Hilbert spaces, thus in equation 23 there is an implicit graded tensor product. Since $p(\check{\mathbf{S}}) = 0$ there will be no sign correction for all operators but $\check{\mathbf{S}}_{L,1}(\theta)$. In fact the single site odd parity elements, which, for $\check{\mathbf{S}}$, can be written as linear combinations of $\sigma^x$ and $\sigma^y$, will be of the form:

$$\sigma_1^\alpha \left( \prod_{i=2}^{L-1} \sigma_i^z \right) \sigma_L^\beta, \quad \alpha, \beta = x, y. \tag{26}$$

These terms under a Jordan-Wigner transformation become nearest neighbour interactions without any sign correction of the form $(-1)^{\hat{N}}$, as previously discussed.

The global fermionic symmetry of the brick wall circuit is inherited from the supersymmetry of the scattering matrix in equation 8. Generalising the supercharges to a $L$ particle space in the following way

$$\mathbf{Q}^{\mathbf{L}}(\gamma, \theta) = \sum_{i=1}^{L/2} \mathbf{Q}^{\mathbf{l}}_{2i-1,2i}(\gamma, \theta), \qquad \mathbf{Q}^{\mathbf{R}}(\gamma, \theta) = \sum_{i=1}^{L/2} \mathbf{Q}^{\mathbf{r}}_{2i-1,2i}(\gamma, \theta), \tag{27}$$

where

$$\mathbf{Q}^{\mathbf{l}}_{2i-1,2i}(\gamma, \theta) = e^{\frac{\gamma+\theta}{4}} \left( \prod_{j<2i-1} \sigma_j^z \right) \sigma_{2i-1}^x + e^{-\frac{\gamma+\theta}{4}} \left( \prod_{j<2i} \sigma_j^z \right) \sigma_{2i}^x, \tag{28}$$

$$\mathbf{Q}^{\mathbf{r}}_{2i-1,2i}(\gamma, \theta) = e^{\frac{\gamma-\theta}{4}} \left( \prod_{j<2i-1} \sigma_j^z \right) \sigma_{2i-1}^y + e^{-\frac{\gamma-\theta}{4}} \left( \prod_{j<2i} \sigma_j^z \right) \sigma_{2i}^y, \tag{29}$$

we find

$$\begin{aligned}
\mathbf{Q}^{\mathbf{L/R}}(\theta)\mathbf{U}_{\mathbf{F}}(\theta) &= \mathbf{Q}^{\mathbf{L/R}}(\theta)\mathbf{U}_{\mathbf{F}}^{\mathbf{o}}(\theta)\mathbf{U}_{\mathbf{F}}^{\mathbf{e}}(\theta), \\
&= \mathbf{U}_{\mathbf{F}}^{\mathbf{o}}(\theta)\mathbf{Q}^{\mathbf{L/R}}(-\theta)\mathbf{U}_{\mathbf{F}}^{\mathbf{e}}(\theta), \\
&= \mathbf{U}_{\mathbf{F}}^{\mathbf{o}}(\theta)\mathbf{U}_{\mathbf{F}}^{\mathbf{e}}(\theta)\mathbf{Q}^{\mathbf{L/R}}(\theta), \\
&= \mathbf{U}_{\mathbf{F}}(\theta)\mathbf{Q}^{\mathbf{L/R}}(\theta).
\end{aligned} \tag{30}$$

The strings of $\sigma^z$ in 29 show the fermionic nature of the operators, each single term in the two sums 27 becomes local after a Jordan-Wigner transformation. A single operator does not commute with the time evolution operator, therefore the support of the two operators $\mathbf{Q}^{\mathbf{L/R}}(\gamma, \theta)$ is the whole chain.

## 2.3 OBC: UBW and fermionic symmetry

A unitary brick wall circuit with open boundary condition (OBC) and global fermionic symmetry can be realized by employing the boundary scattering operators $\mathbf{K}(\theta)$ displayed in eq. 12. However, for the general case with $m_1 \neq m_2$ this will take a circuit with $L$ layers rather than 1 layer.

Let us illustrate this for the case $L = 4$, see also fig. 1b. The natural definition of a UBW circuit with OBC is

$$\mathbf{U}_{\mathbf{F}}^{\text{OBC}}(\alpha, \gamma, \theta) = \left[ \mathbf{K_1}(-\theta/2)\check{\mathbf{S}}_{2,3}(\alpha, \gamma, \theta)\mathbf{K_4}(\theta/2) \right]\left[ \check{\mathbf{S}}_{1,2}(\alpha, \gamma, \theta)\check{\mathbf{S}}_{3,4}(\alpha, \gamma, \theta) \right]. \tag{31}$$

A layer is made of two sublayers (see figure 1b), where in the second sublayer the particles at sites 1 and 4 reflect off the open boundaries. For equal masses, $\gamma = 0$, one quickly checks that $\mathbf{U}_{\mathbf{F}}^{\mathrm{OBC}}(\alpha,\gamma,\theta)$ commutes with $\mathbf{Q}^{\mathbf{L}}(\theta) + \mathbf{Q}^{\mathbf{R}}(\theta)$.

However, for $m_1 \neq m_2$ the operator $\mathbf{U}_{\mathbf{F}}^{\mathrm{OBC}}(\alpha,\gamma,\theta)$ leaves the system in a configuration where sites 1 and 2 have a particle of mass $m_2$ while sites 3 and 4 have a particle of mass $m_1$. On such a state $\mathbf{Q}^{\mathbf{L}}(\gamma,\theta) + \mathbf{Q}^{\mathbf{R}}(\gamma,\theta)$ (which assumes that the particles alternate between $m_1$ and $m_2$ along the chain) is not an appropriate operator. To return to the original configuration of particle masses and rapidities, we need an extended UBW which comprises not 1 but $L = 4$ layers,

$$
\begin{aligned}
\mathbf{U}_{\mathbf{F}}^{\mathrm{OBC,extended}}(\alpha,\gamma,\theta) = {} & \big[\mathbf{K_1}(-\theta/2)\check{\mathbf{S}}_{2,3}(\alpha,\gamma,\theta)\mathbf{K_4}(\theta/2)\big]\big[\check{\mathbf{S}}_{1,2}(\alpha,0,\theta)\check{\mathbf{S}}_{3,4}(\alpha,0,\theta)\big] \\
& \times \big[\mathbf{K_1}(-\theta/2)\check{\mathbf{S}}_{2,3}(\alpha,-\gamma,\theta)\mathbf{K_4}(\theta/2)\big]\big[\check{\mathbf{S}}_{1,2}(\alpha,-\gamma,\theta)\check{\mathbf{S}}_{3,4}(\alpha,-\gamma,\theta)\big] \\
& \times \big[\mathbf{K_1}(-\theta/2)\check{\mathbf{S}}_{2,3}(\alpha,-\gamma,\theta)\mathbf{K_4}(\theta/2)\big]\big[\check{\mathbf{S}}_{1,2}(\alpha,0,\theta)\check{\mathbf{S}}_{3,4}(\alpha,0,\theta)\big] \\
& \times \big[\mathbf{K_1}(-\theta/2)\check{\mathbf{S}}_{2,3}(\alpha,\gamma,\theta)\mathbf{K_4}(\theta/2)\big]\big[\check{\mathbf{S}}_{1,2}(\alpha,\gamma,\theta)\check{\mathbf{S}}_{3,4}(\alpha,\gamma,\theta)\big]. \quad (32)
\end{aligned}
$$

This operator, by construction, commutes with $\mathbf{Q}^{\mathbf{L}}(\gamma,\theta) + \mathbf{Q}^{\mathbf{R}}(\gamma,\theta)$.

## 2.4 Free Fermi form of the scattering matrix

The hidden free fermionic (or matchgate) structure that we discussed in section 1.3 guarantees that the 2-body scattering operator, which furnishes our basic 2-qubit gate, can be written in free fermionic form. We display this form in this section and shall extend it to the many-body UBW in section 4 below. We start by expressing $\check{\mathbf{S}}(\alpha,\gamma,\theta)$ in exponential form

$$
\check{\mathbf{S}}_{i,i+1}(\alpha,\gamma,\theta) = \exp[i\mathbf{E}_{i,i+1}]. \quad (33)
$$

The two-site exponent $\mathbf{E}_{i,i+1}$ can be written as a linear combination of tensor products of Pauli matrices, resulting in

$$
\begin{aligned}
\mathbf{E}_{i,i+1} = {} & \frac{a_{11}}{2}\big(\sigma_i^z + \sigma_{i+1}^z\big) + \frac{a_{12}}{2}\cos(\phi)\big(\sigma_i^x\sigma_{i+1}^x + \sigma_i^y\sigma_{i+1}^y\big) \\
& - \frac{a_{12}}{2}\sin(\phi)\big(\sigma_i^x\sigma_{i+1}^y - \sigma_i^y\sigma_{i+1}^x\big) + \frac{b_{12}}{2}\big(\sigma_i^x\sigma_{i+1}^y + \sigma_i^y\sigma_{i+1}^x\big). \quad (34)
\end{aligned}
$$

The coefficients are found to be the following

$$
\begin{aligned}
a_{11} &= \frac{\sqrt{2}\cosh\left(\frac{\theta}{2}\right)}{\sqrt{2\alpha^2 + \cosh(\theta) + 1}} \arccos\left(\frac{2\alpha\cosh\left(\frac{\gamma}{2}\right)}{\sqrt{2\alpha^2(\cosh(\gamma) + \cosh(\theta)) + \sinh^2(\theta)}}\right), \\
a_{12} &= \arccos\left(\frac{2\alpha\cosh(\frac{\theta}{2})}{\sqrt{2\alpha^2(\cosh(\gamma) + \cosh(\theta)) + \sinh^2(\theta)}}\right), \\
\phi &= \frac{1}{2}\arccos\left(\frac{\sinh^2(\theta) - 4\alpha^2\sinh^2(\frac{\gamma}{2})}{\sinh^2(\theta) + 4\alpha^2\sinh^2(\frac{\gamma}{2})}\right), \\
b_{12} &= \frac{\sqrt{2}\alpha}{\sqrt{2\alpha^2 + \cosh(\theta) + 1}} \arccos\left(\frac{2\alpha\cosh\left(\frac{\gamma}{2}\right)}{\sqrt{2\alpha^2(\cosh(\gamma) + \cosh(\theta)) + \sinh^2(\theta)}}\right).
\end{aligned} \quad (35)
$$

Now we can use the Jordan-Wigner transformation to express our exponent in terms of spinless fermionic operators,

$$
c_i^\dagger = \prod_{j<i}\sigma_j^z\sigma_i^-, \qquad c_i = \prod_{j<i}\sigma_j^z\sigma_i^+, \qquad \sigma_i^z = \mathbb{1} - 2c_i^\dagger c_i. \quad (36)
$$

This leads to

$$\mathbf{E}_{i,i+1} = -a_{11}\left(c_i^\dagger c_i + c_{i+1}^\dagger c_{i+1} - \mathbb{1}\right) + a_{12}\left(e^{i\phi} c_i^\dagger c_{i+1} + e^{-i\phi} c_{i+1}^\dagger c_i\right) + i b_{12}\left(c_i^\dagger c_{i+1}^\dagger - c_{i+1} c_i\right). \quad (37)$$

This form makes clear that we can consider free fermionic 2-qubit gates that venture outside the space parametrized by $\alpha$, $\gamma$, $\theta$, where a global fermionic symmetry is enforced. Using the Jordan-Wigner transformation we can also express the supercharges 29 in terms of spinless fermionic operators as follows:

$$\mathbf{Q^l}_{2i-1,2i}(\gamma,\theta) = e^{\frac{\gamma+\theta}{4}}\left(c_{2i-1}^\dagger + c_{2i-1}\right) + e^{-\frac{\gamma+\theta}{4}}\left(c_{2i}^\dagger + c_{2i}\right), \quad (38)$$

$$\mathbf{Q^r}_{2i-1,2i}(\gamma,\theta) = i e^{\frac{\gamma-\theta}{4}}\left(c_{2i-1}^\dagger - c_{2i-1}\right) + i e^{-\frac{\gamma-\theta}{4}}\left(c_{2i}^\dagger - c_{2i}\right). \quad (39)$$

We will later see that considering perturbations outside the space parametrized by $\alpha$, $\gamma$, $\theta$ breaks the criticality, and give rise to topological phases in the Hamiltonian limit of the UBW.

## 2.5 UBW and Hamiltonian limit

While we loosely think of $\mathbf{U_F}(\alpha,\gamma,\theta)$ as a time evolution operator, with $\theta$ taking the role of time, we should realize that the eigenvalues of $\mathbf{E}_{i,i+1}$ depend on $\theta$ in a non-linear fashion. A standard picture of quantum mechanical time evolution arises in the limit of small $\theta$ and for $\gamma = 0$. Since $\mathbf{U_F}(\alpha,\gamma = 0,\theta = 0)$ is the identity operator, the behaviour of $\mathbf{U_F}$ in this limit is fully captured by its logarithmic derivative, which we call $\mathbf{H_0}(\alpha)$ (section 3.1).

For $\gamma \neq 0$ the $\theta \to 0$ limit of $\mathbf{U_F}(\alpha,\gamma,\theta)$ is not the identity and we should prepend a circuit representing $\mathbf{U_F}(\alpha,\gamma,0)^{-1}$ before we can extract a logarithmic derivative, which we will call $\mathbf{H_\gamma}(\alpha,\gamma)$ (sections 3.2 and 3.3).

The global fermionic symmetry of $\mathbf{U_F}(\alpha,\gamma,\theta)$ carries over to both $\mathbf{H_0}(\alpha)$ and $\mathbf{H_\gamma}(\alpha,\gamma)$. We will find that this implies that these Hamiltonians are critical for all $\alpha$, $\gamma$.

# 3 Brick wall circuit in the Hamiltonian limit

For general $\alpha$, $\gamma$, we can write the following expansion of the Floquet time evolution operator for $\theta \to 0$,

$$\mathbf{U_F}(\theta) = \exp[i\mathbf{E}(\theta)] \stackrel{\theta \to 0}{=} \mathbf{U_F}(0) + i\mathbf{U_F}(0)\mathbf{H_\gamma}\,\theta + o(\theta^2), \quad (40)$$

defining $\mathbf{H_\gamma}$ as the logarithmic derivative of $\mathbf{U_F}(0)^{-1}\mathbf{U_F}(\theta)$ in $\theta = 0$. We first we look at the simpler case with $\gamma = 0$.

## 3.1 $\gamma = 0$

For $\gamma = 0$ the scattering matrix (S-matrix) becomes the identity operator, $\check{\mathbf{S}}(\alpha,0,0) = \mathbb{1}$, thus the only relevant operator will be its first order derivative,

$$\check{\mathbf{S}}(\alpha,0,\theta) \stackrel{\theta \to 0}{=} \mathbb{1} + i \begin{pmatrix} \frac{1}{2\alpha} & 0 & 0 & \frac{-i}{2} \\ 0 & 0 & \frac{1}{2\alpha} & 0 \\ 0 & \frac{1}{2\alpha} & 0 & 0 \\ \frac{i}{2} & 0 & 0 & -\frac{1}{2\alpha} \end{pmatrix} \theta + o(\theta^2). \quad (41)$$

The logarithmic derivative in equation 40 can be analytically derived,

$$\mathbf{H_0}(\alpha) = \frac{\partial}{\partial\theta}\mathbf{E}(\theta)\bigg|_{\theta=0} = \sum_{i=1}^{L/2}\left(\check{\mathbf{S}}'_{2i-1,2i}(\alpha,0,0) + \check{\mathbf{S}}'_{2i,2i+1}(\alpha,0,0)\right). \quad (42)$$

The resulting Hamiltonian depends only on the parameter $\alpha$, and can be written in the spin basis as

$$\mathbf{H_0}(\alpha) = \frac{1}{2\alpha} \sum_{i=1}^{L} \sigma_i^z + \frac{1}{4\alpha} \sum_{i=1}^{L} \left( \sigma_i^x \sigma_{i+1}^x + \sigma_i^y \sigma_{i+1}^y \right) + \frac{1}{4} \sum_{i=1}^{L} \left( \sigma_i^x \sigma_{i+1}^y + \sigma_i^y \sigma_{i+1}^x \right). \tag{43}$$

We remark that the tensor product underlying the Hamiltonian 43 is a graded tensor product, thus the boundary terms of the form $\sigma_L^\alpha \otimes_g \sigma_1^\beta$ are non-local in the spin representation. They take the form $\sigma_L^\alpha \sigma_2^z \ldots \sigma_{L-1}^z \sigma_1^\beta$. It is natural to do a Jordan-Wigner transformation and look at the resulting spinless fermionic chain. Normally using the Jordan-Wigner comes at the cost of a phase when closing the boundary, here that does not happen thanks to the graded tensor product. In fermionic language the Hamiltonian becomes

$$\mathbf{H_0}(\alpha) = \frac{1}{2\alpha} - \frac{1}{\alpha} \sum_{i=1}^{L} c_i^\dagger c_i + \frac{1}{2\alpha} \sum_{i=1}^{L-1} \left( c_i^\dagger c_{i+1} + c_{i+1}^\dagger c_i \right) + \frac{i}{2} \sum_{i=1}^{L-1} \left( c_i^\dagger c_{i+1}^\dagger - c_{i+1} c_i \right). \tag{44}$$

The resulting model is the well known Kitaev chain [32], here realized with chemical potential $\mu = \frac{1}{\alpha}$ and hopping strength $t = \frac{1}{2\alpha}$. Although the Hamiltonian depends on the parameter $\alpha$, it is always critical. Seemingly the only way to break away from criticality is by breaking the global fermionic symmetry inherited from the brick wall circuit. In the Hamiltonian limit the commutation relations eq. 30 become

$$\left[ \mathbf{H_0}(\alpha), \mathbf{Q^L}(0,0) \right] = 0, \qquad \left[ \mathbf{H_0}(\alpha), \mathbf{Q^R}(0,0) \right] = 0. \tag{45}$$

The two operators can be transformed into fermionic operators with a Jordan-Wigner transformation resulting into

$$\mathbf{Q^L}(0,0) = \sum_{i=1}^{L} \gamma_{2i-1}, \qquad \mathbf{Q^R}(0,0) = \sum_{i=1}^{L} \gamma_{2i}. \tag{46}$$

where $\gamma_{2i-1} = c_i^\dagger + c_i$ and $\gamma_{2i} = i(c_i^\dagger - c_i)$ are Majorana fermions.

The two topological phases present in the Kitaev chain are distinguishable on an open chain because in the non-trivial phase there are two unpaired Majorana fermions on the edges. In the critical point these edge modes are delocalized throughout the chain, and they can be identified with the two global fermionic symmetry operators in equation 46. In order to access non-trivial topological phases it is necessary to break the global fermionic symmetry: this leads to the opening of a gap in the spectrum and to the appearance and localization of gapless Majorana modes to the edges of the system (when considering open boundary conditions).

## 3.2 $\gamma \neq 0$ – Spectrum and global fermionic symmetry

Now considering the more general case for $\gamma \neq 0$, the resulting Hamiltonian will be much more complex, since the scattering matrix in $\theta = 0$ is no longer the identity operator,

$$\check{\mathbf{S}}(\alpha, \gamma, \theta) \overset{\theta \to 0}{=} \frac{1}{\cosh\left(\frac{\gamma}{2}\right)} \begin{pmatrix} 1 & 0 & 0 & 0 \\ 0 & 1 & \sinh\left(\frac{\gamma}{2}\right) & 0 \\ 0 & -\sinh\left(\frac{\gamma}{2}\right) & 1 & 0 \\ 0 & 0 & 0 & 1 \end{pmatrix} + \frac{1}{\cosh\left(\frac{\gamma}{2}\right)} \begin{pmatrix} \frac{i}{2\alpha} & 0 & 0 & \frac{1}{2} \\ 0 & 0 & \frac{i}{2\alpha} & 0 \\ 0 & \frac{i}{2\alpha} & 0 & 0 \\ -\frac{1}{2} & 0 & 0 & \frac{-i}{2\alpha} \end{pmatrix} \theta$$
$$+ o(\theta^2). \tag{47}$$

We find the following expression for $\mathbf{H}_\gamma$

$$\mathbf{H}_\gamma = \sum_{i=1}^{L/2} \left[ \check{\mathbf{s}}_{2i-1,2i}^{(0)} \right]^\dagger \check{\mathbf{s}}_{2i-1,2i}^{(1)} + \sum_{i=1}^{L/2} \left[ \check{\mathbf{s}}_{2i-1,2i}^{(0)} \right]^\dagger \left[ \check{\mathbf{s}}_{2i+1,2i+2}^{(0)} \right]^\dagger \left[ \check{\mathbf{s}}_{2i,2i+1}^{(0)} \right]^\dagger \check{\mathbf{s}}_{2i,2i+1}^{(1)} \check{\mathbf{s}}_{2i-1,2i}^{(0)} \check{\mathbf{s}}_{2i+1,2i+2}^{(0)}, \tag{48}$$

where $\check{\mathbf{S}}_{i,i+1}^{(0)} = \check{\mathbf{S}}_{i,i+1}(\alpha, \gamma, 0)$ and $\check{\mathbf{S}}_{i,i+1}^{(1)} = \frac{\partial}{\partial \theta} \check{\mathbf{S}}_{i,i+1}(\alpha, \gamma, \theta)\Big|_{\theta=0}$.

Again applying the Jordan-Wigner transformation 36 we find a free fermionic Hamiltonian which is written explicitly in equation B.1 in Appendix B. Similar Hamiltonians have been studied in recent years [33,34] unveiling a multitude of topological phases. In this subsection we analyze the spectrum of $\mathbf{H}_\gamma$ and its global fermionic symmetry. As for $\gamma = 0$ we will find that the global fermionic symmetry protects criticality and pitches $\mathbf{H}_\gamma$ precisely at a critical surface. This is illustrated in the next subsection, where we add terms breaking the global fermionic symmetry and observe non-trivial topological phases in the BDI class.

Analyzing this Hamiltonian B.1 in momentum space we observe a folding of the Brillouin zone from $[-\pi, \pi]$ to $[-\pi/2, \pi/2]$ and a symmetry for $k \leftrightarrow -k$. The folding, $k \to \pi - k$, is due to the 2-site translational symmetry of the Hamiltonian, while the $k \to -k$ symmetry has its origin in the particle-hole and charge conjugation symmetries. For general momenta $0 < k < \pi/2$ the Hamiltonian takes the form

$$
\begin{aligned}
\mathbf{H}_\gamma{}^k = {} & N_1 \left( c_k^\dagger c_k + c_{-k}^\dagger c_{-k} - 1 \right) + N_2 \left( c_{k-\pi}^\dagger c_{k-\pi} + c_{\pi-k}^\dagger c_{\pi-k} - 1 \right) \\
& + H \left( c_k^\dagger c_{k-\pi} + c_{\pi-k}^\dagger c_{-k} \right) + H^* \left( c_{k-\pi}^\dagger c_k + c_{-k}^\dagger c_{\pi-k} \right) \\
& + S_1 \left( c_k^\dagger c_{-k}^\dagger + c_{\pi-k}^\dagger c_{k-\pi}^\dagger + c_{-k} c_k + c_{k-\pi} c_{\pi-k} \right) \\
& + S_2 \left( c_k^\dagger c_{\pi-k}^\dagger - c_{k-\pi} c_{-k} \right) + (S_2)^* \left( c_{\pi-k} c_k - c_{-k}^\dagger c_{k-\pi}^\dagger \right) .
\end{aligned}
\tag{49}
$$

The explicit expression of the various parameters can be found in appendix B. $\mathbf{H}_\gamma{}^k$ acts on a block of 16 states generated by the 1-fermi operators with momentum $k$, $-k$, $k-\pi$ and $\pi-k$. Two of these states, $|k, k-\pi\rangle$ and $|-k, \pi-k\rangle$ are annihilated by $\mathbf{H}_\gamma{}^k$, there is an irreducible block with 6 states with 0, 2 or 4 particles and there are two $4 \times 4$ blocks with 1 and 3-particle states. Diagonalizing the latter gives dispersion relations of the single-particle (BdG) excitations. We find the dispersions $\pm(\epsilon_\gamma^k)_{1,2}$ with

$$
(\epsilon_\gamma^k)_{1,2} = \frac{1}{\sqrt{2}} \sqrt{v_0 + v_1 \cos(2k) \pm \sqrt{\sum_{j=0}^{6} \mu_j \cos(2jk)}}.
\tag{50}
$$

The coefficients $v_j, \mu_j$ are given in appendix B. Figure 2 shows these dispersion relations for various choices of $\alpha$ and $\gamma$.

The spectrum turns out to be gapless for $k \to 0$, since

$$
(v_0 + v_1)^2 = \sum_{j=0}^{6} \mu_j = \frac{16}{\alpha^4 \cosh^4\left(\frac{\gamma}{2}\right)}.
\tag{51}
$$

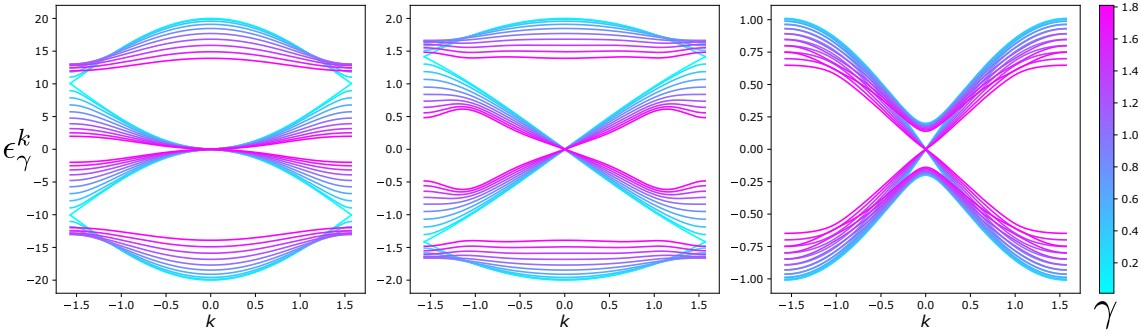

Figure 2: Dispersions $\pm \epsilon_\gamma^k$ for $\mathbf{H}_\gamma$. From the left: $\alpha = 0.1, 1, 10$.

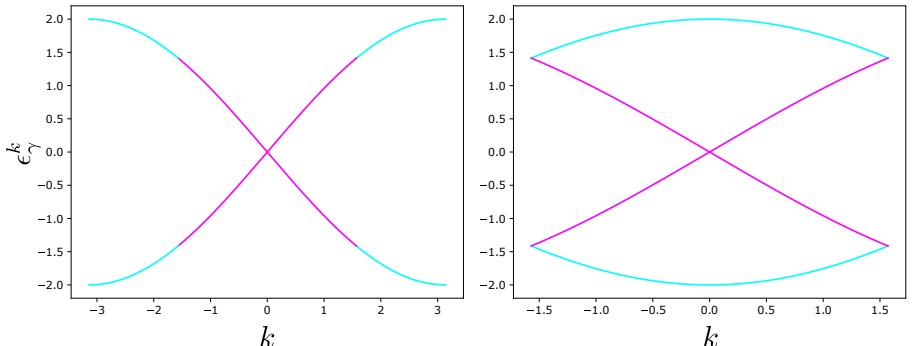

Figure 3: Folding of the Brillouin Zone of Hamiltonian B.1 for $\alpha = 1$, $\gamma = 0$. From the left: $-\pi < k < \pi$, $-\pi/2 < k < \pi/2$.

This implies that $\mathbf{H}_\gamma$ is critical for all $\gamma$ and that topological phases are only possible if the global fermionic symmetry is broken.

To make this explicit, we zoom in on $k = 0$, where the Hamiltonian $\mathbf{H}_\gamma$ reduces to two $2 \times 2$ blocks on the bases $\mathbf{M} = \{|\emptyset\rangle, |0, \pi\rangle\}$ and $\mathbf{N} = \{|0\rangle, |\pi\rangle\}$,

$$\mathbf{H}_\gamma^{\mathbf{0},\mathbf{M}} = \begin{pmatrix} -\frac{N_1^0 + N_2^0}{2} & 0 \\ 0 & \frac{N_1^0 + N_2^0}{2} \end{pmatrix}, \qquad \mathbf{H}_\gamma^{\mathbf{0},\mathbf{N}} = \begin{pmatrix} \frac{N_1^0 - N_2^0}{2} & H^0 \\ H^0 & \frac{-N_1^0 + N_2^0}{2} \end{pmatrix}, \tag{52}$$

with the superscript 0 denoting the value for $k = 0$. Using that

$$N_1^0 = -\frac{2}{\alpha} \frac{\sinh^2(\frac{\gamma}{4})}{\cosh^2(\frac{\gamma}{2})}, \qquad N_2^0 = -\frac{2}{\alpha} \frac{\cosh^2(\frac{\gamma}{4})}{\cosh^2(\frac{\gamma}{2})}, \qquad H^0 = \sqrt{N_1^0 N_2^0}, \tag{53}$$

it is found that both these blocks give energies $\pm(N_1^0 + N_2^0)/2$ and we conclude that the 1-particle (BdG) energies are

$$(\epsilon_\gamma^{k=0})_1 = 0, \quad (\epsilon_\gamma^{k=0})_2 = -N_1^0 - N_2^0 = \frac{2}{\alpha} \frac{1}{\cosh(\frac{\gamma}{2})}. \tag{54}$$

The 1-particle zero-mode going with $(\epsilon_\gamma^{k=0})_1 = 0$ is found to be

$$(\eta_\gamma^{k=0})_1 = \sqrt{\frac{-1}{N_1^0 + N_2^0}} \left( \sqrt{-N_2^0} c_0 - \sqrt{-N_1^0} c_\pi \right) = \frac{1}{2\sqrt{L}} \frac{1}{\cosh(\frac{\gamma}{2})} \left( Q^L(\gamma, 0) - i Q^R(\gamma, 0) \right). \tag{55}$$

This makes explicit that the global fermionic charges constitute a zero mode of $\mathbf{H}_\gamma$ and that both $Q^L(\gamma, 0)$ and $Q^R(\gamma, 0)$ commute with $\mathbf{H}_\gamma$.

In the limit $\gamma \to 0$, where translational symmetry is restored, the model reduces to the critical Kitaev chain. This is shown in figure 3. The model exhibits 4 distinct bands that collapse into 2 in the limit $\gamma \to 0$.

### 3.3 $\gamma \neq 0$ – Symmetry breaking and topological phases

To study the topological properties of the model it is convenient to write the Hamiltonian Bogoliubov-de Gennes (BdG) form,

$$\mathbf{H}_\gamma = \frac{1}{2} \sum_k \Psi_k^\dagger \Lambda_k \Psi_k, \tag{56}$$

with Nambu spinors and local Hamiltonian

$$
\Psi = \begin{pmatrix} c_k \\ c_{k-\pi} \\ c_{-k}^\dagger \\ c_{\pi-k}^\dagger \end{pmatrix}, \qquad \Lambda_k = \begin{pmatrix} N_1 & H & S_1 & S_2 \\ H^* & N_2 & S_2^* & -S_1 \\ S_1 & S_2 & -N_1 & -H \\ S_2^* & -S_1 & -H^* & -N_2 \end{pmatrix}.
\tag{57}
$$

The explicit values of the coefficient can be found in appendix B.

Once a model is written in BdG form, it naturally exhibits particle-hole symmetry. The symmetry operator is then $\mathcal{P} = U_{\mathcal{P}}\mathcal{K}$, where $\mathcal{K}$ is the complex conjugation operator and $U_{\mathcal{P}} = \sigma^x \otimes \mathbb{1}$. It can be checked that $U_{\mathcal{P}}\Lambda_k^* U_{\mathcal{P}}^\dagger = -\Lambda_{-k}$. By looking at the coefficients of the Hamiltonian it might seem that time reversal symmetry is absent, since some have complex values, but that is not the case. The time reversal operator turns out to be $\mathcal{T} = U_{\mathcal{T}}\mathcal{K}$, with $U_{\mathcal{T}} = i\sigma^z \otimes \mathbb{1}$, again the symmetry relation can be checked $U_{\mathcal{T}}\Lambda_k^* U_{\mathcal{T}}^\dagger = \Lambda_{-k}$. We point out that this representation of $\mathcal{T}$ is induced by our conventions yielding a purely imaginary coefficient $\Delta = i b_{12}$ of the superconducting pairing term, see eq. 37. The connection with the more common representation for real $\Delta$, $\mathcal{T} = \mathcal{K}$, can be established through conjugation of $\mathcal{T}, \mathcal{P}$ under the basis rotation $G = e^{i\frac{\pi}{4}(\sigma^z \otimes \mathbb{1})}$, but we stress how these two are totally equivalent. Given that both time reversal and particle hole symmetry are present the Hamiltonian limit of eq. 40 (differently from the more general circuit $\mathbf{U}_{\mathbf{F}}(\alpha, \gamma, \theta)$) exhibits a chiral symmetry with $\mathcal{C} = U_{\mathcal{C}} = U_{\mathcal{P}}U_{\mathcal{T}}$ and $U_{\mathcal{C}}\Lambda_k U_{\mathcal{C}}^\dagger = -\Lambda_k$. Since all three symmetry operators square to the identity, $\mathcal{P}^2 = \mathcal{T}^2 = \mathcal{C}^2 = \mathbb{1}$, the model is in the topological insulator class BDI. Moreover, we point out the absence of sublattice symmetry, caused by the presence of chemical potentials entering as diagonal terms in the representation of our Hamiltonian $\Lambda_k$. This implies the non-existence of SSH-like phases for our model [33].

In $d = 1$ the BDI class is characterized by a $\mathbb{Z}$ index [35, 36]. Given the symmetries of the model and the absence of SSH-like phases, which would manifest through localized fermionic zero-modes, we can only expect the presence of Majorana zero-modes (MZM). In order to probe this, we employ the chiral index [37], whose absolute value quantifies the number of MZM which would localize at each end of an OBC instance of the system. With a unitary rotation we can transform the local Hamiltonian into an off-diagonal operator,

$$
U_{\mathcal{M}} = \frac{1}{2}\begin{pmatrix} 1+i & 0 & 1-i & 0 \\ 0 & 1+i & 0 & 1-i \\ 1+i & 0 & -1+i & 0 \\ 0 & 1+i & 0 & -1+i \end{pmatrix}, \qquad U_{\mathcal{M}}\Lambda_k U_{\mathcal{M}}^\dagger = \begin{pmatrix} 0 & V(k) \\ V^\dagger(k) & 0 \end{pmatrix}.
\tag{58}
$$

Our unitary transformation $U_{\mathcal{M}}$ is related to the usual rotation $U = e^{-i\frac{\pi}{4}(\sigma^y \otimes \mathbb{1})}$ taking a BdG Hamiltonian with real $\Delta$ to a purely off-diagonal form through $G$, as $U_{\mathcal{M}} = UG$. By a redefinition of the rotation $U$ by conjugation under $G$, $V = G^\dagger UG$, one gets, in analogy with [37]:

$$
V\Lambda_k V^\dagger = \begin{pmatrix} 0 & -iV(k) \\ iV^\dagger(k) & 0 \end{pmatrix}.
\tag{59}
$$

Finally, defining the complex function $z(k) = \det[V(k)]/|\det[V(k)]| = \exp[i\psi(k)]$, the winding number is:

$$
W = \frac{1}{2\pi i}\int_{-\frac{\pi}{2}}^{\frac{\pi}{2}} \frac{dz(k)}{z(k)} = \frac{1}{2\pi i}\text{Tr}\int_{-\frac{\pi}{2}}^{\frac{\pi}{2}} dk\, V^{-1}(k)\partial_k V(k).
\tag{60}
$$

The winding number can be explicitly calculated by computing the integral above for an arbitrary choice of the parameters $\alpha$ and $\gamma$; as long as our global fermionic symmetry is in place,

numerical evidence shows that $W = 0$. Therefore the Hamiltonian $\mathbf{H}_\gamma$ describes a topologically trivial model, in agreement with our earlier observation that $\mathbf{H}_\gamma$ is always gapless.

In order to access non-trivial phases we perturb the system such that $\mathcal{C}$, $\mathcal{P}$ and $\mathcal{T}$ symmetries, as well as the free-fermionic nature of the model, are retained, but the global fermionic symmetry is broken. This approach stems from the interpretation of the fermionic symmetry as the responsible for the impossibility to move away from criticality through a change of parameters. In principle, one could choose to perturb directly the UBW generating $\mathbf{H}_\gamma$ through a modification of the coefficients in eq. 35. That would also allow to exit the global fermionic symmetry submanifold, resulting in non-trivial topological phases. Our choice is to extract the Hamiltonian limit from the UBW and then perturb its BdG form directly, leaving the other analysis for future work. We do it by perturbing the coefficients B.2 of $\mathbf{H}_\gamma$:

$$J_{AB}^L \to J_{AB}^L + \delta, \qquad N_A \to N_A + \epsilon_1, \qquad N_B \to N_B + \epsilon_2. \tag{61}$$

Now we can calculate the winding number for fixed values of $\alpha$ and $\gamma$, while changing the strength of the perturbations. As mentioned above, the absolute value of the index $W$ quantifies the number of MZM which can localize at each end of an open chain. Figures 4a-4f show how we are able to access the Kitaev-like phase ($|W| = 1$), which implies localization of a MZM at each end of an OBC instance of the system. In the $\gamma = 0$ limit, shown in fig. 4a, we see how perturbing the chemical potential $\epsilon$ as we keep $\delta = 0$ is equivalent to satisfying the condition $|\mu| < 2|t|$ to achieve the non-trivial topological phase of the Kitaev chain. For $\delta \neq 0$, we observe how the condition is always met in this parameter range as far as $\epsilon < \delta$. We also point out the absence of gap-closing lines corresponding to the boundaries of the $W = 1$ phase in figure 4c. This happens because the $W = \pm 1$ phases differ by a global phase factor and are thus equivalent.

# 4 Brick wall circuits: Spectral analysis

We now turn to the spectral analysis of the full UBW operator $\mathbf{U_F}(\alpha, \gamma, \theta)$. For $\gamma = 0$, the results will reduce to those for the Hamiltonian $\mathbf{H_0}(\alpha)$ in the limit where $\theta \to 0$. For $\gamma \neq 0$ such a direct connection does not exist.

Our analysis will lead to (somewhat implicit) expressions for the 1-particle dispersion relations $\epsilon_i^k$. We anticipate that the global fermionic symmetry that we built into the UBW circuit will force one of the $\epsilon_i^k$ to approach 0 in the limit $k \to 0$, causing a degeneracy in the (logarithmic) spectrum of $\mathbf{U_F}(\alpha, \gamma, \theta)$. Surprisingly, other gapless points arise as well and upon fine-tuning $\theta = \pm\gamma$ we see a coalescence that leads to a cubic dispersion $\epsilon^k \propto k^3$.

## 4.1 Structure of UBW in momentum space

We consider the general UBW $\mathbf{U_F}(\alpha, \gamma, \theta)$ on (an even number of) $L$ sites and with periodic boundary conditions (PBC), see fig. 5. The two sublayers making up $\mathbf{U_F}(\alpha, \gamma, \theta)$ are expressed as

$$\mathbf{U_o} = \exp[i\mathbf{E_o}] = \exp\left[i\sum_{i=1}^{L/2}\mathbf{E}_{2i,2i+1}\right], \qquad \mathbf{U_e} = \exp[i\mathbf{E_e}] = \exp\left[i\sum_{i=1}^{L/2}\mathbf{E}_{2i-1,2i}\right]. \tag{62}$$

We will proceed by writing each layer in momentum space. The explicit breaking of translation symmetry between the even and odd sites causes a reduction of the Brillouin zone to to $-\pi/2 < k \leq \pi/2$. The symmetry $k \to -k$ gives a further reduction and we can analyze the spectral structure restricting to momenta $0 \leq k \leq \pi/2$.

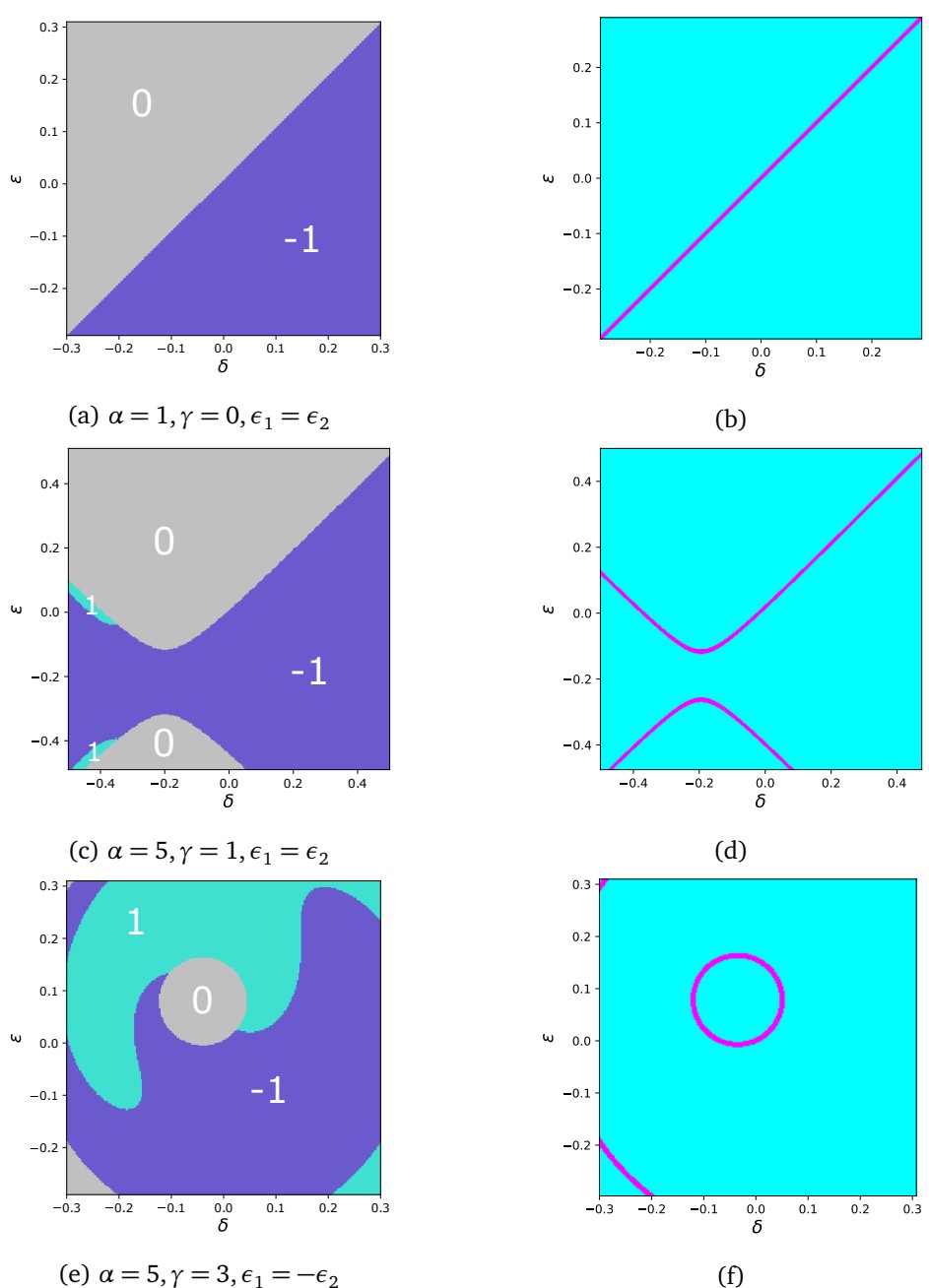

Figure 4: Topological phase diagrams for the chiral index $W$ (left) and energy gap-closing points (right) as functions of $\epsilon$ and $\delta$.

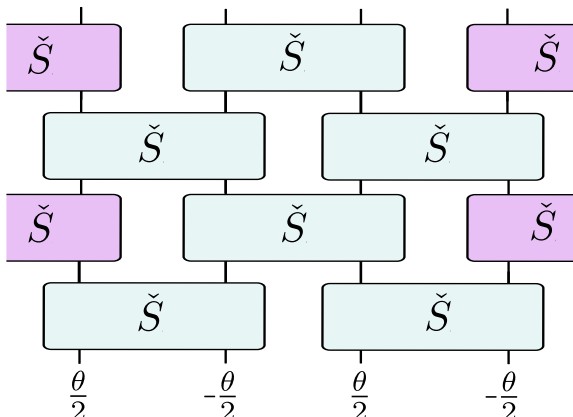

Figure 5: General brick wall circuit with periodic boundary conditions.

For general $k$, the action of the $\mathbf{E}_{i,i+1}$ will mix the 1-fermi operators with momentum $k$, $-k$, $\pi-k$ and $k-\pi$ and we will have to handle the action of the $\mathbf{E}_{i,i+1}$ on the span of the these operators, which in general has dimension 16. We first consider the simpler sectors where $k = 0$ or $k = \pi/2$.

## 4.2 Spectral analysis for $k = 0$ and $k = \pi/2$

For the sector with $k = 0, \pi$ the momentum space exponents read

$$\mathbf{E}_o^0 = -a_{11}\left(c_0^\dagger c_0 + c_\pi^\dagger c_\pi - 1\right) + i b_{12}\left(c_0^\dagger c_\pi^\dagger - c_\pi c_0\right)$$
$$+ a_{12}\cos(\phi)\left(c_0^\dagger c_0 - c_\pi^\dagger c_\pi\right) - i a_{12}\sin(\phi)\left(c_0^\dagger c_\pi - c_\pi^\dagger c_0\right), \tag{63}$$

and

$$\mathbf{E}_e^0 = -a_{11}\left(c_0^\dagger c_0 + c_\pi^\dagger c_\pi - 1\right) - i b_{12}\left(c_0^\dagger c_\pi^\dagger - c_\pi c_0\right)$$
$$+ a_{12}\cos(\phi)\left(c_0^\dagger c_0 - c_\pi^\dagger c_\pi\right) + i a_{12}\sin(\phi)\left(c_0^\dagger c_\pi - c_\pi^\dagger c_0\right). \tag{64}$$

This gives rise to two $2 \times 2$ blocks for the exponents in matrix form. On the basis $\{|\emptyset\rangle, |0,\pi\rangle\}$ (with $|\emptyset\rangle$ denoting the state annihilated by $c_0$ and $c_\pi$ and $|0,\pi\rangle = c_0^\dagger c_\pi^\dagger |\emptyset\rangle$) these are

$$\mathbf{E}_o^{0,\mathbf{M}} = \begin{pmatrix} a_{11} & -i b_{12} \\ +i b_{12} & -a_{11} \end{pmatrix}, \qquad \mathbf{E}_e^{0,\mathbf{M}} = \begin{pmatrix} a_{11} & i b_{12} \\ -i b_{12} & -a_{11} \end{pmatrix}, \tag{65}$$

and on basis $\{|0\rangle, |\pi\rangle\}$ (with $|0\rangle = c_0^\dagger |\emptyset\rangle$ and $|\pi\rangle = c_\pi^\dagger |\emptyset\rangle$))

$$\mathbf{E}_o^{0,\mathbf{N}} = a_{12}\begin{pmatrix} \cos\phi & -i\sin\phi \\ i\sin\phi & -\cos\phi \end{pmatrix}, \qquad \mathbf{E}_e^{0,\mathbf{N}} = a_{12}\begin{pmatrix} \cos\phi & i\sin\phi \\ -i\sin\phi & -\cos\phi \end{pmatrix}. \tag{66}$$

It is an elementary exercise to exponentiate and then multiply these matrices and to extract the characteristic polynomial $\mathbf{Char}^0(x) = \prod(\lambda_i - x)$ of $\mathbf{U}_F^0(\alpha, \gamma, \theta)$ in both these sectors.

Specializing to $a_{11}$, $a_{12}$, $\phi$ and $b_{12}$ as given in eq. 35, we obtain, in both the $M$ and the $N$ sectors,

$$\mathbf{Char}^0(x) = x^2 - 2\frac{2\alpha^2(\cosh(\gamma) + \cosh(\theta)) - \sinh(\theta)^2}{2\alpha^2(\cosh(\gamma) + \cosh(\theta)) + \sinh(\theta)^2}x + 1. \tag{67}$$

Clearly, the global fermionic symmetry maps between the two sectors and causes the eigenvalues to be identical between the two. Unitarity and particle-hole symmetry guarantee that the eigenvalues come in a pair $\{\lambda, \lambda^*\}$ with $|\lambda| = 1$.

Translating to 1-particle energies, we find

$$\epsilon_1^{k=0} = 0, \qquad \epsilon_2^{k=0} = 2\arccos\left(\frac{2\alpha^2(\cosh(\gamma)+\cosh(\theta))-\sinh^2(\theta)}{2\alpha^2(\cosh(\gamma)+\cosh(\theta))+\sinh^2(\theta)}\right), \tag{68}$$

and we have $\mathbf{U_F^0}(\alpha,\gamma,\theta) = \exp[i\sum_j \epsilon_j^{k=0}(\eta_j^\dagger\eta_j - \frac{1}{2})]$. The 1-particle operators for the zero-energy mode are given by linear combinations of the two fermionic charges,

$$\eta_1^{k=0} = \frac{1}{2\sqrt{L}}\left(\frac{\mathbf{Q^L}(\gamma,\theta)}{\sqrt{\cosh(\frac{\gamma+\theta}{2})}} - i\frac{\mathbf{Q^R}(\gamma,\theta)}{\sqrt{\cosh(\frac{\gamma-\theta}{2})}}\right). \tag{69}$$

Repeating the analysis for the sector with $k = \frac{\pi}{2}$ and $k = -\frac{\pi}{2}$, we find on basis $\left\{|\emptyset\rangle, |\frac{\pi}{2}, -\frac{\pi}{2}\rangle\right\}$

$$\mathbf{E_o^{\pi/2,M}} = \begin{pmatrix} a_{11} & b_{12} \\ b_{12} & -a_{11} \end{pmatrix}, \qquad \mathbf{E_e^{\pi/2,M}} = \begin{pmatrix} a_{11} & b_{12} \\ b_{12} & -a_{11} \end{pmatrix}, \tag{70}$$

and on basis $\{|\frac{\pi}{2}\rangle, |-\frac{\pi}{2}\rangle\}$

$$\mathbf{E_o^{\pi/2,N}} = a_{12}\begin{pmatrix} \sin\phi & i\cos\phi \\ -i\cos\phi & -\sin\phi \end{pmatrix}, \qquad \mathbf{E_e^{\pi/2,N}} = a_{12}\begin{pmatrix} \sin\phi & -i\cos\phi \\ i\cos\phi & -\sin\phi \end{pmatrix}. \tag{71}$$

This leads to

$$\mathbf{Char}^{\pi/2,N}(x) = x^2 - 2\left(2 - \frac{8\alpha^2(\cosh(\gamma)-1)}{2\alpha^2(\cosh(\gamma)+\cosh(\theta))+\sinh^2(\theta)}\right)x + 1, \tag{72}$$

and

$$\mathbf{Char}^{\pi/2,M}(x) = x^2 - 2\left(-2 + \frac{8\alpha^2(\cosh(\gamma)+1)}{2\alpha^2(\cosh(\gamma)+\cosh(\theta))+\sinh^2(\theta)}\right)x + 1. \tag{73}$$

In this case the 1-particle energies are both non-zero in general. However, one of the 1-particle energies $\epsilon_i^{k=\pi/2}$ vanishes when the sectors $\mathbf{M}$, $\mathbf{N}$ give identical eigenvalues, which happens for

$$2\alpha^2\big(\cosh(\gamma)-\cosh(\theta)\big) = \sinh^2(\theta). \tag{74}$$

## 4.3 Spectral analysis for general k

For fixed $k$ satisfying $0 < k < \pi/2$ we find the following expressions

$$\begin{aligned}
\mathbf{E_{o,e}^k} = \big\{ &-a_{11}\left(c_k^\dagger c_k + c_{-k}^\dagger c_{-k} + c_{k-\pi}^\dagger c_{k-\pi} + c_{\pi-k}^\dagger c_{\pi-k} - 2\right) \\
&+ a_{12}\cos(k-\phi)\left(c_k^\dagger c_k - c_{k-\pi}^\dagger c_{k-\pi}\right) \\
&+ a_{12}\cos(k+\phi)\left(c_{-k}^\dagger c_{-k} - c_{\pi-k}^\dagger c_{\pi-k}\right) \\
&\pm ia_{12}\sin(k-\phi)\left(c_k^\dagger c_{k-\pi} - c_{k-\pi}^\dagger c_k\right) \\
&\pm ia_{12}\sin(k+\phi)\left(c_{\pi-k}^\dagger c_{-k} - c_{-k}^\dagger c_{\pi-k}\right) \\
&\mp ib_{12}\cos(k)\left(c_k^\dagger c_{\pi-k}^\dagger - c_{\pi-k}c_k + c_{-k}^\dagger c_{k-\pi}^\dagger - c_{k-\pi}c_{-k}\right) \\
&+ b_{12}\sin(k)\left(c_k^\dagger c_{-k}^\dagger + c_{-k}c_k + c_{\pi-k}^\dagger c_{k-\pi}^\dagger + c_{k-\pi}c_{\pi-k}\right)\big\},
\end{aligned} \tag{75}$$

with the top (bottom) sign referring to the odd (even) layer.

The block structure for given $k$ is similar to that for $\mathbf{H}_\gamma$, described in section 3.2. The exponents $\mathbf{E_o^k}$ and $\mathbf{E_e^k}$ decompose in blocks of dimension 1, 4, 6, 4, 1. The corresponding bases are

1. $M_1$: $\{|k, k+\pi\rangle\}$ and $M_1'$: $\{|-k, -k+\pi\rangle\}$. On both these states both the even and odd exponents act trivially, meaning these states are inert under the UBW operator,

2. $M_6$: $\{|\emptyset\rangle, |k, -k\rangle, |k, -k+\pi\rangle, |k+\pi, -k\rangle, |k+\pi, -k+\pi\rangle, |k, k+\pi, -k, -k+\pi\rangle\}$,

3. $N_4$: $\{|k\rangle, |k+\pi\rangle, |k, k+\pi, -k\rangle, |k, k+\pi, -k+\pi\rangle\}$,

4. $N_4'$: $\{|-k\rangle, |-k+\pi\rangle, |-k, -k+\pi, k\rangle, |-k, -k+\pi, k+\pi\rangle\}$.

The unitary brick wall (UBW) operator in this sector takes the form

$$\mathbf{U_F^k}(\alpha, \gamma, \theta) = \exp\left[i \sum_i \epsilon_i^k \left(\eta_i^{k\dagger}\eta_i^k - \frac{1}{2}\right)\right]. \tag{76}$$

This implies that the 16 eigenvalues in the sector labeled with $k$ are products of four factors of the form $\lambda_i = \exp(i\epsilon_i^k)$ or $\lambda_i^* = \exp(-i\epsilon_i^k)$, $i = 1, 2, 3, 4$. Knowing that the states $M_1$ and $M_1'$ have zero energy we conclude that some linear combination of the energies $\epsilon_i$ vanishes, say $\pm(\epsilon_1 + \epsilon_2 + \epsilon_3 + \epsilon_4) = 0$. In fact, for $\gamma = 0$ we have $\epsilon_1 + \epsilon_3 = 0$ and $\epsilon_2 + \epsilon_4 = 0$.

Clearly, the states in the sector $N_4$ and $N_4'$ are related to the state $M_1$ by a single creation or annihilation operator, hence a single operator $\eta_i^\dagger$. The single particle energies $\epsilon_i$ are thus found by diagonalizing $N_4$ and $N_4'$. The characteristic polynomial of $\mathbf{U_F^k}$ on the basis $N_4$ takes the form

$$\begin{aligned}\mathbf{Char}^{k,N_4}(x) &= (x-\lambda_1)(x-\lambda_2)(x-\lambda_3)(x-\lambda_4)\\ &= (x-\lambda_1)(x-\lambda_2)(x-\lambda_3)(x-\lambda_1^*\lambda_2^*\lambda_3^*),\end{aligned} \tag{77}$$

and it will thus be of the general form

$$\mathbf{Char}^{k,N_4}(x) = x^4 - a_4 x^3 + b_4 x^2 - a_4^* x + 1. \tag{78}$$

The other $4 \times 4$ block will have a similar form but with the conjugate coefficients.

$$\mathbf{Char}^{k,N_4'}(x) = x^4 - a_4^* x^3 + b_4 x^2 - a_4 x + 1. \tag{79}$$

Clearly, via all definitions made in the above, the coefficients $a_4$, $b_4$ can be expressed in the parameters $\alpha$, $\gamma$ and $\theta$ of the defining 2-qubit gate $\mathbf{\check{S}}(\alpha, \gamma, \theta)$ but the derivation of these expressions is quite cumbersome. In the most general case the two coefficients are

$$\begin{aligned}a_4 = {}&\frac{1}{2\left(2\alpha^2(\cosh(\gamma)+\cosh(\theta))+\sinh^2(\theta)\right)^2}\\ &\times\Big(8\alpha^2\big[8\alpha^2\cosh(\gamma)\cosh(\theta)+8\alpha^2+\cosh(\theta)-\cosh(3\theta)\big]\\ &\quad+\big[16\alpha^4(\cosh(2\gamma)+\cosh(2\theta))-32\alpha^4-32\alpha^2\cosh(\gamma)\sinh^2(\theta)\\ &\quad\quad-4\cosh(2\theta)+\cosh(4\theta)+3\big]\cos(2k)\\ &\quad-64i\,\alpha^2\sinh(\gamma)\sinh^2(\theta)\sin(2k)\Big),\end{aligned} \tag{80}$$

$$\begin{aligned}b_4 = {}&\frac{1}{2\left(2\alpha^2(\cosh(\gamma)+\cosh(\theta))+\sinh^2(\theta)\right)^2}\\ &\times\Big(16\alpha^4(\cosh(2\gamma)+\cosh(2\theta))+160\alpha^4-96\alpha^2\cosh(\gamma)\sinh^2(\theta)\\ &\quad-4\cosh(2\theta)+\cosh(4\theta)+3\\ &\quad+64\alpha^2\big[-2\alpha^2+\cosh(\theta)(2\alpha^2\cosh(\gamma)-\sinh^2(\theta))\big]\cos(2k)\\ &\quad+\big[4\alpha^2(\cosh(\gamma)-\cosh(\theta))+2\sinh^2(\theta)\big]^2\cos(4k)\Big).\end{aligned} \tag{81}$$

A useful check is that in the limit $k \to 0$ the solutions to $\mathbf{Char}^{k,N_4}(x) = 0$ reduce to $\exp[\pm i\epsilon_1^{k=0}] = 1$ and $\exp[\pm i\epsilon_2^{k=0}]$. Another check is that $a_4 = 4$ and $b_4 = 6$ in the limit $\gamma, \theta \to 0$, as all eigenvalues reduce to 1 in that limit.

In the general case, where $a_4^* \neq a_4$, solving for $\lambda_i^k$ requires solving a quartic equation. However, we observe that the numbers $c_{12} = \cos(\epsilon_1^k + \epsilon_2^k)$, $c_{13} = \cos(\epsilon_1^k + \epsilon_3^k)$ and $c_{23} = \cos(\epsilon_2^k + \epsilon_3^k)$ satisfy

$$c_{12} + c_{13} + c_{23} = \frac{b_4}{2}, \qquad c_{12}c_{13} + c_{12}c_{23} + c_{13}c_{23} = \frac{a_4 a_4^* - 4}{4}, \qquad c_{12}c_{13}c_{23} = \frac{a_4^2 + (a_4^*)^2 - 4b_4}{8}.$$

This implies that the $c_{ij}$ can be expressed as the roots of a third order polynomial, giving closed-form but involved expressions for the $\epsilon_i^k$.

In the following subsections we briefly report on the special limits where $\gamma = 0$ (equal masses) or $\gamma = \theta$.

### 4.3.1 Equal masses

First the case with equal masses, $\gamma = 0$, so that also $\phi = 0$. In this case $\mathbf{Char}^{k,N_4}(x) = \mathbf{Char}^{k,N_4'}(x)$, $a_4$ is real and the expressions for $a_4$ and $b_4$ reduce to

$$
\begin{aligned}
a_4 = \frac{-1}{(-1 + 2\alpha^2 + \cosh(\theta))^2} \\
\times \Bigg( \frac{4\alpha^2}{\cosh^2(\theta/2)} [1 - 4\alpha^2 - 2\cosh(\theta) + \cosh(2\theta)] \\
- 2\tanh^2(\theta/2)[-1 - 8\alpha^2 + 8\alpha^4 + \cosh(2\theta)]\cos(2k) \Bigg),
\end{aligned}
\tag{82}
$$

$$
\begin{aligned}
b_4 = \frac{1}{(-1 + 2\alpha^2 + \cosh(\theta))^2} \\
\times \Bigg( \frac{1}{8\cosh^4(\theta/2)} [3 + 48\alpha^2 + 176\alpha^4 + 4(-1 + 4\alpha^2(-3 + \alpha^2))\cosh(2\theta) + \cosh(4\theta)] \\
- 8\alpha^2 \frac{\sinh^2(\theta/2)}{\cosh^4(\theta/2)} [1 - 4\alpha^2 + 2\cosh(\theta) + \cosh(2\theta)]\cos(2k) \\
+ 2\tanh^4(\theta/2)[1 - 2\alpha^2 + \cosh^2(\theta)]^2 \cos(4k) \Bigg).
\end{aligned}
\tag{83}
$$

For $\gamma = 0$ the cosines of the (logarithmic) energies $\epsilon_i^k$ can be expressed as

$$\cos(\epsilon_{1,2}^k) = \frac{a_4}{4} \pm \frac{1}{4}\sqrt{a_4^2 - 4b_4 + 8},\tag{84}$$

bringing us as close as possible to a closed form expression.

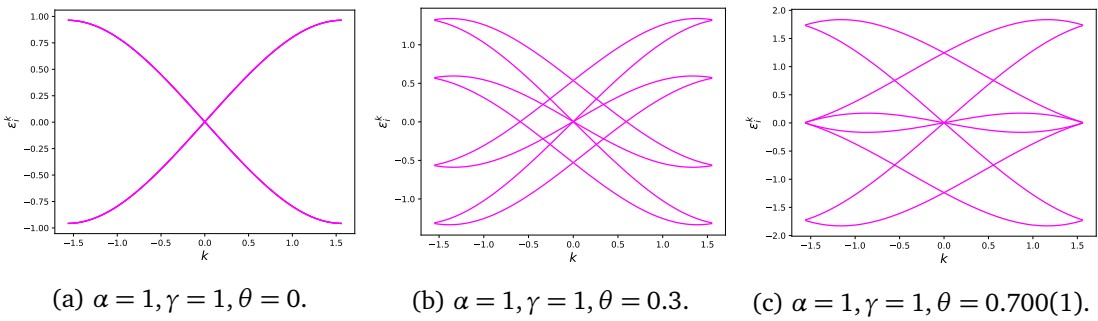

(a) $\alpha = 1, \gamma = 1, \theta = 0$.  (b) $\alpha = 1, \gamma = 1, \theta = 0.3$.  (c) $\alpha = 1, \gamma = 1, \theta = 0.700(1)$.

Figure 6: $\mathbf{U_F}$ spectrum for $\theta \leq \theta_c = 0.700109$.

### 4.3.2 The case $\gamma = \theta$

Note that in this case $a_{12}^2 = a_{11}^2 + b_{12}^2$. The coefficients of the characteristic polynomial of the UBW on basis $N_4$ now come out as

$$
\begin{aligned}
a_4 = \ & \frac{1}{(4\alpha^2 \cosh(\theta) + \sinh^2(\theta))^2} \\
& \times \Big( 4\alpha^2 (\cosh(\theta) + 4\alpha^2 (3 + \cosh(2\theta)) - \cosh(3\theta)) \\
& \quad + 2\sinh^2(\theta)(-1 + 16\alpha^4 - 8\alpha^2 \cosh(\theta) + \cosh(2\theta)) \cos(2k) \\
& \quad - 32\, i\, \alpha^2 \sinh^3(\theta) \sin(2k) \Big),
\end{aligned}
\tag{85}
$$

$$
\begin{aligned}
b_4 = \ & \frac{1}{2} \frac{1}{(4\alpha^2 \cosh(\theta) + \sinh^2(\theta))^2} \\
& \times \Big( 3 + 160\alpha^4 - 4\cosh(2\theta) + 8\alpha^2 (3\cosh(\theta) + 4\alpha^2 \cosh(2\theta) - 3\cosh(3\theta)) \\
& \quad + \cosh(4\theta) - 64\alpha^2 (-2\alpha^2 + \cosh(\theta)) \sinh^2(\theta) \cos(2k) \\
& \quad + 4\sinh^4(\theta) \cos(4k) \Big).
\end{aligned}
\tag{86}
$$

Specializing to $k \to 0$, it is quickly checked that there are two solutions converging on $\epsilon = 0$. Analyzing their dispersion, we find

$$
\epsilon_1^{k \to 0} = -2\tanh(\theta)k + \mathcal{O}(k^2), \qquad \epsilon_2^{k \to 0} = \frac{\sinh(\theta)}{8\alpha^2} k^3 + \mathcal{O}(k^4).
\tag{87}
$$

The cubic dispersion that we find here is unique to the choice $\theta = \pm\gamma$.

### 4.4 Dispersion relations for $U_F$

We illustrate the spectral structure of $\mathbf{U_F}$ by showing some plots (figure 6,7,8) of the 1-particle dispersion $\pm\epsilon_i^k$, $i = 1, \ldots, 4$ and $k \in [0, \pi/2]$.

As soon as $\theta > 0$, there are two branches $\epsilon_1^k$ and $\epsilon_2^k$ that are gapless at $k = 0$, while a third branch crosses zero at a finite $k$. For a critical value $\theta_c$ this finite $k$ reaches the value $k = \pi/2$. The critical value $\theta_c[\alpha, \gamma]$ is precisely the gapless point at $k = \pi/2$ that we identified in eq. 74. For $\alpha = 1$, $\gamma = 1$ we find $\theta_c = 0.700109$.

Increasing $\theta$ beyond $\theta_c$, the values $\pm k$ giving a vanishing dispersion move towards $k = 0$ and precisely for $\theta = \gamma$ they merge into a single multi-critical point with cubic dispersion, see eq. 87.

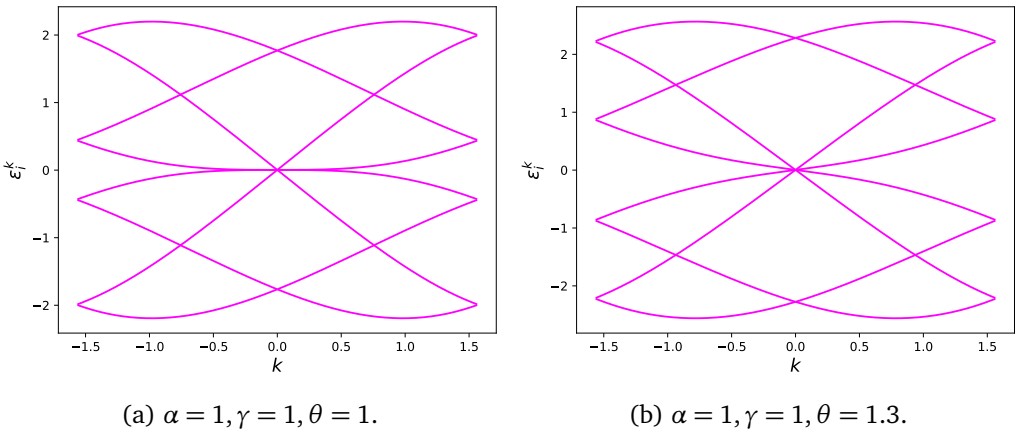

(a) $\alpha = 1, \gamma = 1, \theta = 1$.

(b) $\alpha = 1, \gamma = 1, \theta = 1.3$.

Figure 7: $\mathbf{U_F}$ spectrum for $\theta \geq \theta_c$.

We next consider these same dispersion plots for more general parameter choices which break the global fermionic symmetry. One way to break this symmetry is to scale the amplitude $a_{12}$ with a factor of $t_a$ with respect to the value, given in eq. 35, required by the fermionic symmetry. Clearly, reducing $t_a$ below $t_a = 1$ is analogous to setting $2t < \mu$ in the Kitaev chain eq. 44 and one expects that a gap will open up.

Inspecting the behavior, we see that, for $\theta < \theta_c$ both the perturbations $t_a < 1$ and $t_a > 1$ move the gapless points away from $k = 0$, but do not open a gap for all $k$. Only for $\theta \geq \theta_c$ and $t_a$ sufficiently below $t_a = 1$ does an overall gap open up. Once $\theta \geq \gamma$, an arbitrary perturbation $t_a < 1$ suffices to gap out all branches of the dispersion. This is also shown in figure 8.

Changing the values of $\alpha$ and $\gamma$ gives a similar picture (as long as both remain non-zero). The critical value $\theta_c$ depends on both $\alpha$ and $\gamma$, while the point giving rise to a cubic dispersion is $\theta = \pm\gamma$ for all values of $\alpha$. We stress that the dispersion relations discussed in this section do not pertain to the Hamiltonian $\mathbf{H}_\gamma$, but rather to the exponent $\mathbf{E}(\theta)$ characterizing the brick wall unitary $\mathbf{U_F}$ at finite values of $\theta$, see eq. 15.

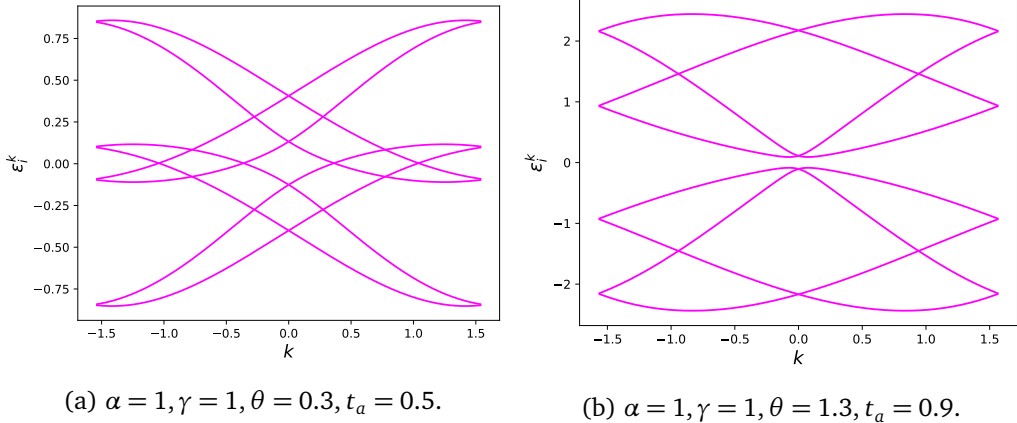

(a) $\alpha = 1, \gamma = 1, \theta = 0.3, t_a = 0.5$.

(b) $\alpha = 1, \gamma = 1, \theta = 1.3, t_a = 0.9$.

Figure 8: $\mathbf{U_F}$ spectrum without global fermionic symmetry.

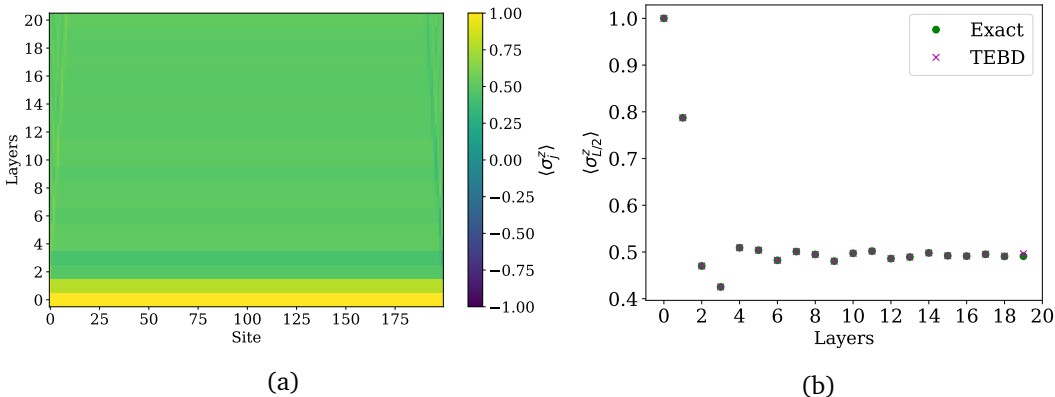

$$\text{(a)} \qquad\qquad\qquad\qquad \text{(b)}$$

Figure 9: (a) $\langle \sigma_j^z \rangle$ TEBD results for an OBC system with $L = 200$, $\alpha = 1$, $\gamma = 0$, $\theta = 0.5$ (b) Local magnetization $\langle \sigma_j^z \rangle$ with $j = L/2$ as a function of number applied $\mathbf{U_F}$ layers, exact results vs. TEBD.

## 5 Simulations of dynamics

This section presents a first exploration of (quench) dynamics generated by applying $N_l$ layers of our brick wall unitary $\mathbf{U_F}$ on some initial state. In principle, these dynamics are tractable via the underlying free fermionic structure, but this analysis quickly becomes cumbersome. We therefore resort to numerics, employing a time-evolving block decimation (TEBD) algorithm [38,39]. We focus on the expectation value of the local magnetization $\sigma_i^z = (1 - 2n_i)$, with $n_i = c_i^\dagger c_i$, after applying $N_l$ layers of $\mathbf{U_F}$ on our initial state, leaving other observables and quantities such as entanglement entropies for later study.

In our numerical analysis, we consider an OBC system which we probe in its bulk, with interest in computing the expectation value of the observables in the pre-thermalization phase, before the onset of finite size effects (features propagating in from the boundary).

In appendix C we complement this analysis with analytical reasoning for the system with PBC, employing the free fermionic spectral structure discussed in section 4. For sufficiently short timescales $N_l$, these results exactly reproduce bulk features studied by our TEBD numerics for OBC. We fix the bond dimension of the TEBD algorithm to $D = 75$ for all the simulations reported below.

### 5.1 Initial state $|00...00\rangle$

We compute the local magnetizations $\langle \sigma_j^z \rangle$ as a function of the number $N_l$ of applied layers $\mathbf{U_F}$ on the all-0 initial state. In figure 9 we show the magnetizations (panel a) and display the evolution of $\sigma_{L/2}^z$ as a function of the number $N_l$ of applied layers (panel b), comparing the TEBD results with exact values, for $\mathbf{U_F}$ with $\alpha = 1$, $\gamma = 0$, $\theta = 0.5$. Appendix C discusses the derivation of the exact (analytical) results, employing the spectral structure for a circuit with PBC.

The evolution of the $\sigma_j^z$ shows a clear equilibration, caused by a dephasing of the contributions of different momentum sectors. It persists until finite-size effects, causing a rephasing, kick in. In appendix C we demonstrate how the equilibrium value of $\sigma_j^z$ can be reproduced on the basis of a generalized Gibbs ensemble (GGE), with the occupation numbers $n_i^k$, $i = 1, \ldots, 4$ of the free fermionic eigenmodes $\eta_i^k$ acting as the conserved quantities constraining the dynamics. Using this GGE, the equilibrium value of the $\sigma_j^z$ is expressed in the expectations values of $n_i^k$ in the initial state.

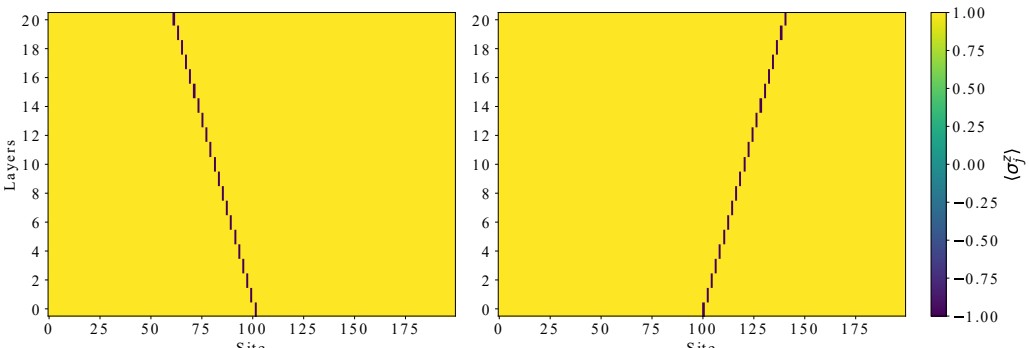

Figure 10: TEBD simulation of the propagation of a seed starting from an odd (even) site for $L = 200$, $\alpha = 1$, $\gamma = 10$, $\theta = 1$.

We remark that for $\gamma > 0$ the equilibrium values of $\sigma_j^z$ depend on the parity of $j$ modulo 2, as is apparent in our figure 11 below. Both equilibrium values can be extracted from the GGE, see appendix C for the details.

## 5.2 Initial state $|00...1...00\rangle$

We next consider an initial state with a single seed '1' in the background of the all-0 state.

For $\gamma = 0$ (corresponding to equal particle masses in the scattering matrix $\check{\mathbf{S}}$) we observe a left-right symmetric response of the magnetizations $\sigma_j^z$ within a cone spanned by velocities $\pm v_{\max}$, whose magnitude depends on $\alpha$ and $\theta$. For $\alpha$ and $\theta$ such that $|f(\theta)| \ll |g(\theta)|$ (see eq. 10), so that the graded permutation term dominates in $\check{\mathbf{S}}$, the magnitude approaches the maximal value $|v_{\max}| = 2$ set by the geometry of the circuit.

The response is markedly different for $\gamma > 0$. In that case, we see a clear 'ballistic' propagation of the seed '1', with drift velocity $v_d$. The sign of $v_d$ depends on the parity modulo 2 of the location of the seed in the initial state. The effect is most pronounced for $\gamma \gg \theta$ and values of $\alpha$ large enough to avoid dominance of the graded permutation term in $\check{\mathbf{S}}$. See figure 10 for the case with $\gamma = 10$, $\theta = 1$, showing ballistic propagation with velocities $v_d$ close to $\pm 2$.

For intermediate values of $\gamma$, the ballistic propagation is still clearly visible. See figure 11, where we display the propagation of the magnetization for $\alpha = 1$ and $\gamma = \theta = 1$ and $\gamma = \theta = 3$, with the initial seed at an even site for both panels.

While the precise value of $v_d$ is not directly tractable from the free fermionic spectral structure (it builds up as an average $\frac{\partial}{\partial k}\epsilon_i^k$ over all participating free fermionic modes), we can track

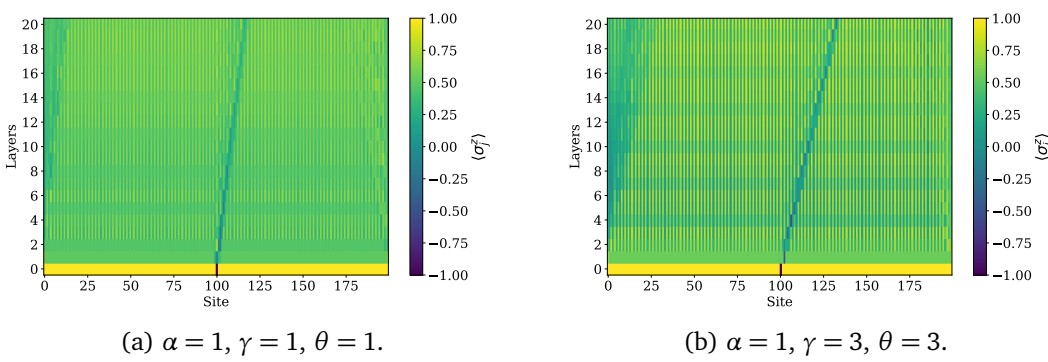

(a) $\alpha = 1$, $\gamma = 1$, $\theta = 1$.         (b) $\alpha = 1$, $\gamma = 3$, $\theta = 3$.

Figure 11: $v_d$ for $\alpha = 1$ and two choices of $\gamma = \theta$.

its value in the limit where $\gamma \gg \theta$ and $\alpha$ large enough to avoid dominance of the graded perturbation term in $\check{\mathbf{S}}$. In that situation, the largest velocity in the system is that coming from one of the gapless branches, with value $v_+ = 2\tanh((\gamma + \theta)/2)$ (see appendix C). For $\gamma \gg \theta$ this velocity is close to the value

$$v_d = 2\tanh(\gamma/2) = 2\frac{m_1 - m_2}{m_1 + m_2}, \tag{88}$$

which we already quoted in our introduction, eq. 1, and which bears a remarkable resemblance to the analogous formula eq. 2 reported in [11].

## 6 Outlook

In this paper we embarked on the analysis of a class of unitary brick wall quantum circuits with a constituent 2-qubit gate $\check{\mathbf{S}}(\alpha, \gamma, \theta)$ that is of free fermionic form. The operator $\check{\mathbf{S}}$ was earlier found and studied as a scattering matrix in specific integrable supersymmetric particle theories in 1+1D. The supersymmetry of these particle theories translates into a global fermionic symmetry of the circuit unitaries $\mathbf{U_F}$. We found that the global fermionic symmetry protects the criticality of two Hamiltonian operators associated to $\mathbf{U_F}$: the 'Floquet' Hamiltonian $\mathbf{E}(\alpha, \gamma, \theta)$ and the logarithmic derivative $\mathbf{H}_\gamma(\alpha, \gamma)$ (see eq. 40). The latter simplifies to the Kitaev chain Hamiltonian $\mathbf{H_0}(\alpha)$ for $\gamma = 0$. We identified topological phases of $\mathbf{H}_\gamma$ (in the BDI class) that arise upon breaking the global fermionic symmetry (but keeping the free fermionic form). The (ground state) phase diagrams of (perturbations of) $\mathbf{H}_\gamma(\alpha, \gamma)$ deserve further study. Perturbations may include terms beyond the free fermionic realm, similar to those considered for the Kitaev chain in [40, 41].

We also explored the quench dynamics of $\mathbf{U_F}(\alpha, \gamma, \theta)$, focusing on the time evolution of the polarizations $\sigma_j^z$ starting from the all-0 state or from a state with a single seed. Our numerics revealed two main features. One is the equilibration of polarizations $\sigma_j^z$ to an average value which can be reproduced by a GGE based on the free fermionic structure. The other is the manifestation of a drift velocity $v_d$, which can be linked to the spectral structure we unveiled in section 4. We intend to follow up on these results, including other observables such as entanglement entropies and aiming to get a better grasp of the (non-local) 'Floquet' Hamiltonian $\mathbf{E}(\alpha, \gamma, \theta)$ and the associated dynamics.

## Acknowledgments

Many thanks to Vladimir Gritsev for discussions and guidance in the early stages of this project. We also thank Philippe Corboz, Bruno Bertini and Tomaž Prosen for discussions and Kun Zhang for alerting us to the connection with matchgates. We thank the SciPost referees for their constructive criticisms.

**Funding information**   This work was supported by the Dutch Ministry of Economic Affairs and Climate Policy (EZK), as part of the Quantum Delta NL programme. We thank the Rudolf Peierls Centre for Theoretical Physics at the University of Oxford, where part of this work was done, for hospitality and financial support through EPSRC grant EP/N01930X/1.

# A   Graded Floquet Baxterization

In this appendix we generalize the Floquet Baxterization thereom of [31] to the case of a graded space, assuming periodic boundary conditions. We adhere to the notation of [31], naming the solution to the Yang-Baxter equation $\check{\mathbf{R}}$ rather than $\check{\mathbf{S}}$.

**Theorem A.1** (Graded Floquet Baxterization). *For a unitary and invertible $\check{\mathbf{R}}$-matrix in a $\mathbb{Z}_2$ graded space such that $p(\check{\mathbf{R}}(u)) = 0$, the periodic time evolution operator of a system of size $L$, with $L \mod 2 = 0$ and $\check{\mathbf{R}}(u) = \mathbf{\Pi}\mathbf{R}(u)$:*

$$\hat{\mathbf{U}}_\mathbf{F}(\theta_1 - \theta_2) = \left(\prod_{i=1}^{L/2} \check{\mathbf{R}}_{2i-1,2i}(\theta_1 - \theta_2)\right)\left(\prod_{i=1}^{L/2} \check{\mathbf{R}}_{2i,2i-1}(\theta_1 - \theta_2)\right), \tag{A.1}$$

*is integrable, i.e.*

$$\left[\hat{\mathbf{U}}_\mathbf{F}(\theta_1, \theta_2), \mathbf{t}(u, \theta_1, \theta_2)\right] = 0, \qquad \left[\mathbf{t}(u, \theta_1, \theta_2), \mathbf{t}(v, \theta_1, \theta_2)\right] = 0, \tag{A.2}$$

*with the transfer matrix evaluated with:*

$$\mathbf{t}(u, \theta_1, \theta_2) = \mathrm{Tr}^s\left[\mathbf{T}_\mathbf{a}(u, \theta_1, \theta_2)\right], \qquad \mathbf{T}_\mathbf{a}(u, \theta_1, \theta_2) = \prod_{i=L}^{1} \mathbf{R}_{a,i}(u, \theta_j), \quad j = i \mod 2. \tag{A.3}$$

We remark that the monodromy matrix in equation A.3 is made up by **R**-matrices and since each one of them acts both on the auxiliary space $\mathcal{H}_a$ and one physical qubit $\mathcal{H}_i$ their form is not trivial. In fact even if $p(\mathbf{R}(u)) = 0$ when only one of the two subspaces is permuted if $\mathbf{R}(u)$ has non diagonal terms there will be extra minus signs, from equation 19. Explicitly the product will be written as [29]:

$$(\mathbf{T}_\mathbf{a}(u, \theta_1, \theta_2))_{\alpha\mathbf{a}}^{\beta\mathbf{b}} = \left(\mathrm{Tr}^s_a\left[\prod_{i=L}^{1} \mathbf{R}_{a,i}(u - \theta_j)\right]\right)_{\alpha\mathbf{a}}^{\beta\mathbf{b}} (-1)^{\sum_{j=2}^{L}(p(a_j)+p(b_j))\sum_{i=1}^{j-1} p(a_i)}. \tag{A.4}$$

*Proof.* We start by defining the operator $\mathbf{W}(\theta_1, \theta_2)$ acting on the graded Hilbert space $\left(\mathbb{C}^{(1|1)}\right)^{\otimes L}$:

$$\mathbf{W}(\theta_1, \theta_2) = \tilde{\mathbf{G}} \prod_{i=1}^{L/2} \check{\mathbf{R}}_{2m-1,2m}(\theta_1, \theta_2), \qquad \tilde{\mathbf{G}} = \prod_m^{L-1} \mathbf{\Pi}_{m,m+1}, \tag{A.5}$$

where $\tilde{\mathbf{G}}$ is the graded translation operator. We now want to introduce an operator $\tilde{\mathbf{W}}$ that acts on a bigger Hilbert space $\mathcal{H}_b \otimes \left(\mathbb{C}^{(1|1)}\right)^{\otimes L}$, to define **W** as we define the transfer matrix with the monodromy matrix:

$$\tilde{\mathbf{W}}_b(u_1, u_2) = \prod_{i=L/2}^{1} \mathbf{R}_{b,2m}(\theta_1, \theta_2)\mathbf{\Pi}_{b,2m-1}. \tag{A.6}$$

Knowing that $\mathbf{R}_{i,j} = \mathbf{\Pi}_{i,j}\check{\mathbf{R}}_{i,j}$ we get for each term of the product:

$$\begin{aligned}
\mathbf{R}_{b,2m}\mathbf{\Pi}_{b,2m-1} &= \mathbf{\Pi}_{b,2m}\check{\mathbf{R}}_{b,2m}\mathbf{\Pi}_{b,2m-1} \\
&= \mathbf{\Pi}_{b,2m}\mathbf{\Pi}_{b,2m-1}\mathbf{\Pi}_{b,2m-1}\check{\mathbf{R}}_{b,2m-1}\mathbf{\Pi}_{b,2m-1} \\
&= \mathbf{\Pi}_{b,2m}\mathbf{\Pi}_{b,2m-1}\check{\mathbf{R}}_{2m-1,2m}.
\end{aligned} \tag{A.7}$$

So plugging in this result in the equation for each pair of **R** and **Π**, we get:

$$\tilde{\mathbf{W}}_b(\theta_1, \theta_2) = \prod_{i=L}^{1} \mathbf{\Pi}_{b,i} \prod_{i=1}^{L/2} \check{\mathbf{R}}_{2m-1,2m}(\theta_1, \theta_2).$$ (A.8)

Now using the train trick A.7 we can rewrite the previous expression as:

$$\tilde{\mathbf{W}}_b(\theta_1, \theta_2) = \mathbf{\Pi}_{b,1} \tilde{\mathbf{G}} \prod_{i=1}^{L/2} \check{\mathbf{R}}_{2m-1,2m}(\theta_1, \theta_2).$$ (A.9)

We underline that the order of the product of the bricks doesn't matter, since they act on different subspaces, thus we get the relation:

$$\mathbf{W}(\theta_1, \theta_2) = \mathrm{Tr}^s_{\,b} \left[ \tilde{\mathbf{W}}_b(\theta_1, \theta_2) \right].$$ (A.10)

Now the key relation that makes the system integrable is an equation similar to Yang-Baxter. In fact one can show that for a graded **R**-matrix,

$$\mathbf{R}_{a,b}(u, \theta_1) \mathbf{R}_{a,m}(u, \theta_1) \mathbf{\Pi}_{b,m} = \mathbf{\Pi}_{b,m} \mathbf{R}_{a,m}(u, \theta_1) \mathbf{R}_{a,b}(u, \theta_1).$$ (A.11)

Therefore one can reproduce the intertwining relation between inhomogeneous monodromy matrices [42] with the newly introduced operator $\tilde{\mathbf{W}}_b(\theta_1, \theta_2)$,

$$\mathbf{R}_{a,b}(u, \theta_1) \mathbf{M}_{a,m}(u, \theta_1, \theta_2) \tilde{\mathbf{W}}_b(\theta_1, \theta_2) = \tilde{\mathbf{W}}_b(\theta_1, \theta_2) \mathbf{M}_{a,m}(u, \theta_1, \theta_2) \mathbf{R}_{a,b}(u, \theta_1).$$ (A.12)

Now multiplying this last equation from the left and taking the super partial trace on both sides one get:

$$[\mathbf{T}(u, \theta_1, \theta_2), \mathbf{W}(\theta_1, \theta_2)] = 0.$$ (A.13)

From construction the transfer matrix commutes with the graded translation operator squared,

$$\left[ \mathbf{T}(u, \theta_1, \theta_2), \tilde{\mathbf{G}}^2 \right] = 0,$$ (A.14)

and naturally also with its inverse $\tilde{\mathbf{G}}^{-2}$. The square of the operator now we notice that the operator $\mathbf{W}(\theta_1, \theta_2)$ is essentially the even layer of the brick wall multiplied by the graded translation operator. Taking the square of it and multiplying it from the left by $\tilde{\mathbf{G}}^{-2}$ will give us:

$$\begin{aligned}
\tilde{\mathbf{G}}^{-2} \mathbf{W}^2(\theta_1, \theta_2) &= \tilde{\mathbf{G}}^{-1} \left( \prod_{i=1}^{L/2} \check{\mathbf{R}}_{2m-1,2m}(\theta_1, \theta_2) \right) \tilde{\mathbf{G}} \left( \prod_{i=1}^{L/2} \check{\mathbf{R}}_{2m-1,2m}(\theta_1, \theta_2) \right) \\
&= \left( \prod_{i=1}^{L/2} \check{\mathbf{R}}_{2m,2m+1}(\theta_1, \theta_2) \right) \left( \prod_{i=1}^{L/2} \check{\mathbf{R}}_{2m-1,2m}(\theta_1, \theta_2) \right) \\
&= \mathbf{U}_{\mathbf{F}}(\theta_1, \theta_2).
\end{aligned}$$ (A.15)

Therefore,

$$[\mathbf{T}(u, \theta_1, \theta_2), \mathbf{U}_{\mathbf{F}}(\theta_1, \theta_2)] = 0, \quad \forall u \in \mathbb{C}.$$ (A.16)

$\square$

We remark that in the theorem we imposed the condition $p(\mathbf{R}(u)) = 0$ because when this condition does not hold we have to deal with the minus signs from equation 19. That would mean that the representation in ungraded space of the **R**-matrix would not be local anymore. This would make the implementation of the quantum brick wall circuit unfeasible.

# B  Explicit form of $H_\gamma$

## B.1  Real space form and coefficients

The Hamiltonian in equation 48 can be expressed as a linear combination of local Pauli matrices. Writing it in block diagonal form and then applying a Jordan-Wigner transformation (36) the real space Hamiltonian results:

$$
\begin{aligned}
\mathbf{H}_\gamma = \sum_i \Bigg[ & -N_A c_{2i-1}^\dagger c_{2i-1} - N_B c_{2i}^\dagger c_{2i} + J_{AB}\left(c_{2i-1}^\dagger c_{2i} + c_{2i}^\dagger c_{2i-1}\right) \\
& + J_{BA}\left(c_{2i}^\dagger c_{2i+1} + c_{2i+1}^\dagger c_{2i}\right) + J_{AB}^L\left(c_{2i-1}^\dagger c_{2i+2} + c_{2i+2}^\dagger c_{2i-1}\right) \\
& + J_{AA}\left(c_{2i-1}^\dagger c_{2i+1} + c_{2i+1}^\dagger c_{2i-1} - c_{2i}^\dagger c_{2i+2} - c_{2i+2}^\dagger c_{2i}\right) \\
& + iS_{AB}\left(c_{2i-1}^\dagger c_{2i}^\dagger - c_{2i}c_{2i-1}\right) + iS_{BA}\left(c_{2i}^\dagger c_{2i+1}^\dagger - c_{2i+1}c_{2i}\right) \\
& + iS_{AA}\left(c_{2i-1}^\dagger c_{2i+1}^\dagger - c_{2i+1}c_{2i-1} - c_{2i}^\dagger c_{2i+2}^\dagger + c_{2i+2}c_{2i}\right) \\
& + S_{AB}^L\left(c_{2i-1}^\dagger c_{2i+2}^\dagger + c_{2i+2}c_{2i-1}\right) \Bigg] + \frac{L}{4}(N_A + N_B).
\end{aligned}
\tag{B.1}
$$

This Hamiltonian is clearly free fermionic and its coefficient are functions of the parameters $\alpha$ and $\gamma$, explicitly:

$$
\begin{aligned}
N_A &= \frac{\operatorname{sech}\left(\frac{\gamma}{2}\right) - \tanh\left(\frac{\gamma}{2}\right)\operatorname{sech}^3\left(\frac{\gamma}{2}\right)}{\alpha}, &
N_B &= \frac{\operatorname{sech}\left(\frac{\gamma}{2}\right) + \tanh\left(\frac{\gamma}{2}\right)\operatorname{sech}^3\left(\frac{\gamma}{2}\right)}{\alpha}, \\
J_{AB} &= \frac{3\tanh^2\left(\frac{\gamma}{2}\right)\operatorname{sech}^2\left(\frac{\gamma}{2}\right) + \operatorname{sech}^4\left(\frac{\gamma}{2}\right)}{2\alpha}, &
J_{BA} &= \frac{\operatorname{sech}^4\left(\frac{\gamma}{2}\right)}{2\alpha}, \\
J_{AA} &= \frac{\tanh\left(\frac{\gamma}{2}\right)\operatorname{sech}^3\left(\frac{\gamma}{2}\right)}{2\alpha}, &
J_{AB}^L &= -\frac{\tanh^2\left(\frac{\gamma}{2}\right)\operatorname{sech}^2\left(\frac{\gamma}{2}\right)}{2\alpha}, \\
S_{AB} &= \frac{\operatorname{sech}\left(\frac{\gamma}{2}\right)}{2}, &
S_{BA} &= \frac{\operatorname{sech}^3\left(\frac{\gamma}{2}\right)}{2}, \\
S_{AA} &= \frac{\tanh\left(\frac{\gamma}{2}\right)\operatorname{sech}^2\left(\frac{\gamma}{2}\right)}{2}, &
S_{AB}^L &= -\frac{\tanh^2\left(\frac{\gamma}{2}\right)\operatorname{sech}\left(\frac{\gamma}{2}\right)}{2}.
\end{aligned}
\tag{B.2}
$$

## B.2  Momentum space and dispersion relation coefficients

The Hamiltonian 57 is the Fourier transform of equation B.1 written after a double folding of the Brillouin zone, its coefficients are:

$$
\begin{aligned}
N_1 &= \frac{\operatorname{sech}^4\left(\frac{\gamma}{2}\right)\left(-4\cosh^3\left(\frac{\gamma}{2}\right) + (3\cosh(\gamma)+1)\cos(k) - 2\sinh^2\left(\frac{\gamma}{2}\right)\cos(3k)\right)}{4\alpha}, \\
N_2 &= \frac{\operatorname{sech}^4\left(\frac{\gamma}{2}\right)\left(-4\cosh^3\left(\frac{\gamma}{2}\right) - (3\cosh(\gamma)+1)\cos(k) + 2\sinh^2\left(\frac{\gamma}{2}\right)\cos(3k)\right)}{4\alpha}, \\
H &= -\frac{\tanh\left(\frac{\gamma}{2}\right)\operatorname{sech}^3\left(\frac{\gamma}{2}\right)\left(\sinh^2\left(\frac{\gamma}{2}\right) + 2i\sinh\left(\frac{\gamma}{2}\right)\sin^3(k) + \cos(2k)\right)}{\alpha}, \\
S_1 &= -\operatorname{sech}^3\left(\frac{\gamma}{2}\right)\sin(k)\left(1 - \sinh^2\left(\frac{\gamma}{2}\right)\cos(2k)\right), \\
S_2 &= 2\sinh\left(\frac{\gamma}{2}\right)\operatorname{sech}^3\left(\frac{\gamma}{2}\right)\sin(k)\cos(k)\left(1 + i\sinh\left(\frac{\gamma}{2}\right)\sin(k)\right).
\end{aligned}
\tag{B.3}
$$

The diagonalization of the two $4 \times 4$ blocks in terms of the coefficients appearing above results in:

$$
\epsilon = \pm \left( \frac{N_1^2}{2} + \frac{N_2^2}{2} + |H|^2 + S_1^2 + |S_2|^2 \pm \frac{1}{2} \left( (N_1^2 - N_2^2)^2 + 4(N_1 + N_2)^2 |H|^2 \right. \right.
$$
$$
+ 16 S_1^2 |H|^2 - 16(N_1 - N_2) S_1 \mathrm{Re}\left[ S_2 H^* \right] - 8 \mathrm{Re}\left[ S_2^2 (H^*)^2 \right]
$$
$$
\left. \left. + 4(N_1 - N_2)^2 |S_2|^2 + 8 |H|^2 |S_2|^2 \right)^{\frac{1}{2}} \right)^{\frac{1}{2}}. \tag{B.4}
$$

Now plugging inside the equation above the values of the coefficients we find the expression in equation 50, with the following coefficients

$$
\nu_0 = \frac{\mathrm{sech}^4\left(\frac{\gamma}{2}\right)}{\alpha^2} \left( 1 + 2\cosh(\gamma) \right) + \mathrm{sech}^2\left(\frac{\gamma}{2}\right),
$$
$$
\nu_1 = \frac{\mathrm{sech}^4\left(\frac{\gamma}{2}\right)}{\alpha^2} - \mathrm{sech}^2\left(\frac{\gamma}{2}\right),
$$
$$
\mu_0 = \frac{\mathrm{sech}^6\left(\frac{\gamma}{2}\right)}{\alpha^4} \left( 8\cosh(\gamma) \right)
$$
$$
+ \frac{\tanh^2\left(\frac{\gamma}{2}\right)\mathrm{sech}^{12}\left(\frac{\gamma}{2}\right)}{32\alpha^2} \left( 300 - 193\cosh(\gamma) + 162\cosh(2\gamma) - 15\cosh(3\gamma) + 2\cosh(4\gamma) \right),
$$
$$
\mu_1 = \frac{8\,\mathrm{sech}^6\left(\frac{\gamma}{2}\right)}{\alpha^4}
$$
$$
- \frac{\tanh^2\left(\frac{\gamma}{2}\right)\mathrm{sech}^{12}\left(\frac{\gamma}{2}\right)}{16\alpha^2} \left( 163 - 120\cosh(\gamma) + 92\cosh(2\gamma) - 8\cosh(3\gamma) + \cosh(4\gamma) \right),
$$
$$
\mu_2 = \frac{\tanh^4\left(\frac{\gamma}{2}\right)\mathrm{sech}^{10}\left(\frac{\gamma}{2}\right)}{16\alpha^2} \left( 93 + 4\cosh(\gamma) + 31\cosh(2\gamma) \right),
$$
$$
\mu_3 = -\frac{\tanh^4\left(\frac{\gamma}{2}\right)\mathrm{sech}^{10}\left(\frac{\gamma}{2}\right)}{2\alpha^2} \left( 21 - 12\cosh(\gamma) + 7\cosh(2\gamma) \right),
$$
$$
\mu_4 = \frac{\tanh^4\left(\frac{\gamma}{2}\right)\mathrm{sech}^{10}\left(\frac{\gamma}{2}\right)}{8\alpha^2} \left( 45 - 44\cosh(\gamma) + 15\cosh(2\gamma) \right),
$$
$$
\mu_5 = -\frac{4\tanh^8\left(\frac{\gamma}{2}\right)\mathrm{sech}^6\left(\frac{\gamma}{2}\right)}{\alpha^2},
$$
$$
\mu_6 = \frac{\tanh^8\left(\frac{\gamma}{2}\right)\mathrm{sech}^6\left(\frac{\gamma}{2}\right)}{2\alpha^2}. \tag{B.5}
$$

## C  Dynamics of $U_F$ with PBC

Exploiting the spectral structure worked out in section 4, we can analyze the dynamics of $\mathbf{U_F}$ with periodic boundary conditions (PBC) in closed form. In this appendix we give a quantitative account of the state produced by $N_l$ layers of $\mathbf{U_F}$ on the all-0 initial state, and we give a more qualitative discussion of the drift velocity $v_d$ of a seed '1' in a background of an all-0 state.

### C.1  Creation and annihilation operators

We assume PBC and $L = 4l + 2$. The momentum sectors are then grouped as $(k = 0, \pi)$ and $(k, -k, k + \pi, -k + \pi)$ for $k = \frac{2\pi j}{L}$, $l = 1, \dots, l$.

Let $v_1, \ldots, v_4$ be the (real-valued) eigenvectors of $\mathbf{U_F^k}$ on the basis $N_4$ (see section 4.3), denoted by $\mathbf{U_F^{k|N4}}$, for momentum $0 < k < \pi/2$. We have

$$v_i^1 |k\rangle + v_i^2 |k+\pi\rangle + v_i^3 |k, k+\pi, -k\rangle + v_i^4 |k, k+\pi, -k+\pi\rangle = \eta_i^{k\dagger} |k, k+\pi\rangle, \qquad \text{(C.1)}$$

with

$$\eta_i^{k\dagger} = \left[ -v_i^1 c_{k+\pi} + v_i^2 c_k + v_i^3 c_{-k}^\dagger + v_i^4 c_{-k+\pi}^\dagger \right]. \qquad \text{(C.2)}$$

Note that

$$\eta_i^k |k, k+\pi\rangle = 0. \qquad \text{(C.3)}$$

Let $V^k$ be the matrix with rows $v_i$, then

$$(V^k)^T V^k = 1, \qquad (V^k)^T \Lambda^k V^k = \mathbf{U_F^{k|N4}}, \qquad \text{(C.4)}$$

with $\Lambda^k$ diagonal with elements $\Lambda_{jj}^k = e^{i\epsilon_j^k}$.

Similarly, on the basis $N_4'$, the operators $\eta_i^k$ create eigenstates acting on $|-k, -k+\pi\rangle$,

$$\eta_i^k = \left[ -\bar{v}_i^1 c_{-k+\pi} + \bar{v}_i^2 c_{-k} + \bar{v}_i^3 c_k^\dagger + \bar{v}_i^4 c_{k+\pi}^\dagger \right], \qquad \text{(C.5)}$$

with $\bar{v}_i$ the eigenvectors of $\mathbf{U_F^k}$ on the basis $N_4'$. We have

$$\overline{V^k} = V^{-k} = V^k M, \qquad \text{(C.6)}$$

with $M$ a 4 by 4 matrix with nonzero entries $M_{14} = M_{41} = -1$, $M_{23} = M_{32} = 1$.

## C.2  Quench dynamics from the all-0 state

In momentum space the initial state $|00\ldots0\rangle$ is annihilated by all operators $c_k^\dagger$, meaning that it is a product over all $k$ values, with $0 \le k < \pi/2$ of the state we denoted as $|\emptyset\rangle$ in section 4. For $0 < k < \pi/2$ this state is part of the basis $M_6$. The restriction of $\mathbf{U_F}$ to $M_6$, denoted as $\mathbf{U_F^{k|M6}}$ is a $6 \times 6$ matrix whose structure is specified in section 4.3. The contributions of a sector with momentum $0 < k < \pi/2$ to the sums $N_{\text{even}}$ and $N_{\text{odd}}$ of the densities $n_j$ on all even (odd) sites are found to be the expectation values of

$$(c_k^\dagger \pm c_{k+\pi}^\dagger)(c_k \pm c_{k+\pi})/2 + (c_{-k}^\dagger \pm c_{-k+\pi}^\dagger)(c_{-k} \pm c_{-k+\pi})/2, \qquad \text{(C.7)}$$

in the state generated by $N_l$ times the action of $\mathbf{U_F}$. This leads to

$$N_{\text{even}}^k = (v_6^k)^T D_{\text{even}} v_6^k, \qquad N_{\text{odd}}^k = (v_6^k)^T D_{\text{odd}} v_6^k, \qquad \text{(C.8)}$$

with $v_6^k = (\mathbf{U_6^{k|M6}})^{N_l} |\emptyset\rangle$ and

$$D_{\text{even}} = \begin{pmatrix} 0 & 0 & 0 & 0 & 0 & 0 \\ 0 & 1 & \frac{1}{2} & \frac{1}{2} & 0 & 0 \\ 0 & \frac{1}{2} & 1 & 0 & \frac{1}{2} & 0 \\ 0 & \frac{1}{2} & 0 & 1 & \frac{1}{2} & 0 \\ 0 & 0 & \frac{1}{2} & \frac{1}{2} & 1 & 0 \\ 0 & 0 & 0 & 0 & 0 & 2 \end{pmatrix}, \qquad D_{\text{odd}} = \begin{pmatrix} 0 & 0 & 0 & 0 & 0 & 0 \\ 0 & 1 & -\frac{1}{2} & -\frac{1}{2} & 0 & 0 \\ 0 & -\frac{1}{2} & 1 & 0 & -\frac{1}{2} & 0 \\ 0 & -\frac{1}{2} & 0 & 1 & -\frac{1}{2} & 0 \\ 0 & 0 & -\frac{1}{2} & -\frac{1}{2} & 1 & 0 \\ 0 & 0 & 0 & 0 & 0 & 2 \end{pmatrix}. \qquad \text{(C.9)}$$

There is a similar (but simpler) expression for the contribution from the sector with momenta $0$ and $\pi$. Summing over all momenta leads to $N_{\text{even}}$ and $N_{\text{odd}}$ and the expectation values of $\sigma_j^z$ for a single even or odd site are simply obtained as

$$\langle \sigma_{\text{even}}^z \rangle = 1 - \frac{4N_{\text{even}}}{L}, \qquad \langle \sigma_{\text{odd}}^z \rangle = 1 - \frac{4N_{\text{odd}}}{L}. \qquad \text{(C.10)}$$

For $\gamma = 0$ the even and odd expectation values are the same. Figure 9(b) displays $\langle \sigma_{\text{even}}^z \rangle$ for the case with $\alpha = 1$, $\gamma = 0$, $\theta = 0.5$.

### C.3 Equilibrium values from GGE

The expectation values $\bar{n}_i^k = \langle \eta_i^{k\dagger} \eta_i^k \rangle$ are all conserved quantities. This allows us to conceive a generalized Gibbs ensemble (GGE) for the quench dynamics.

Applying $N_l$ times $\mathbf{U_F}$ to the all-0 state leads to a state with rapidly oscillating phase factors. Combining all $k$ we will see a dephasing, which is constrained by the conservation of the $n_i^k$. In the resulting equilibrium state we have $\langle \eta_i^{k\dagger} \eta_j^k \rangle = \delta_{ij} \bar{n}_i^k$. This relation allows us to express the expectation values of expressions such as eq. C.7 in terms of $v_i^j$ and $\bar{n}_i^k$. In this way, the ensemble averages for $N_{\text{even}}^k$ and $N_{\text{odd}}^k$, and thereby for the even and odd magnetizations, are fully expressed in data pertaining to $\mathbf{U_F}$ and the initial state. We checked that these ensemble averages agree with the equilibrium values generated by the full dynamics.

### C.4 Velocities of free fermionic modes

The dynamics of an initial state with a single seed '1' reveals characteristic velocities, which can be related to the group velocities $\frac{\partial}{\partial k} \epsilon_i^k$ of the free fermionic modes $\eta_i^k$. While the relations expressing these velocities in terms of our parameters $\alpha$, $\gamma$ and $\theta$ are in general intractable analytically, we found closed-form expressions for momenta approaching 0 or $\pi/2$.

Near momentum $k = 0$ we have

1. four critical branches with $\epsilon_i^{k=0} = 0$ and velocities $\pm v_+$ and $\pm v_-$, with

$$v_+ = 2\tanh((\gamma + \theta)/2), \quad v_- = 2\tanh((\gamma - \theta)/2), \tag{C.11}$$

2. four branches with (in general) non-zero $\epsilon_i^{k=0}$ and velocities $\pm v_0$,

$$v_0 = (v_+ + v_-)/2 = 2\frac{\sinh(\gamma)}{\cosh(\gamma) + \cosh(\theta)}. \tag{C.12}$$

Note that the dispersions for $\theta = \gamma$, as displayed in eq. 87, are a special case where $v_- = 0$ and the leading term in the dispersion of the $v_-$ branch is cubic in $k$.

Near momentum $k = \pi/2$ all velocities turn out to be equal to $\pm v_1$,

$$v_1 = \frac{2\sinh(\theta)}{\sqrt{-1 + 2\alpha^2 + \cosh(\theta)}\sqrt{1 + 2\alpha^2 + \cosh(\theta)}}. \tag{C.13}$$

We note that, as soon as $\theta > 0$, $\lim_{\alpha \to 0} v_1 = 2$, which is the maximal velocity set by the geometry of the system. In a large part of the $\alpha$, $\gamma$, $\theta$ parameter space $v_+$ is the largest velocity in the system. These velocities provide upper bounds for the drift velocity $v_d$. In the particular case where $\gamma \gg \theta$ and $\alpha$ sufficiently large, the drift velocity is found to be close to $v_+$, which in turn is close to the value displayed in eq. 1.

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
