# Peer review of "Brick Wall Quantum Circuits with Global Fermionic Symmetry"

_SciPost Physics, doi:SciPost Phys. 17, 087 (2024)_

## Round 1 · Referee Report · Anonymous (Referee 1) · 2024-7-19

Report

The authors have satisfactorily addressed the comments made in my previous report. I recommend the present version fur publication.

Recommendation

Publish (meets expectations and criteria for this Journal)

---

## Round 1 · Referee Report · Anonymous (Referee 2) · 2024-8-2

Report

The Authors have considerably improved the manuscript. I also find section 5 to be more meaningful now.

There are some typos remaining and there is some room for improvement of readability in section 2.1: please see my remarks below. I suggest another careful check for typos and maybe adding a sentence or two more in section 2.1, then the paper can be published.

Requested changes

(1) The line immediately below eq. 1.2: In order for the notation for spins to be consistent with the one used later on, one should probably change s1 -> s_1 and s2 -> s_2 in "… lines carry SU(2) representations (s1,s2)."

(2) Regarding eq. 1.13: I would still suggest the Authors to mention how K(\theta) is extended to act on two-particle states, since Q=Q^l + Q^r is a two-particle operator.

(3) Section 2.1: I still find this section a bit cryptic. Specifically, the Authors should explain what is the purpose of computational-basis vectors that act as indices of the local operator \hat{O}_i. One thing that is not clear from the narrative is the following: if left-hand side of eq. (2.4) is a matrix element, then why the right-hand side is not? This section should be written in a way that makes that clear.

Recommendation

Ask for minor revision

---

## Round 1 · Referee Report · Anonymous (Referee 3) · 2024-9-3

Report

The Authors have considerably improved the clarity of the paper.
Moreover, they replaced the confusing entanglement entropy section with a clear section on evolution of local magnetisation.

The only small change that should be considered is to enlarge the fontisize of the numerical plot. Each tick/label/legend entry should have the same size of the caption at least.

Apart from this, the manuscript should be published in Scipost Physics.

Requested changes

  • Enlarge fontlabel of figures to at least match that of the captions.

Recommendation

Publish (easily meets expectations and criteria for this Journal; among top 50%)

---

## Round 1 · Author Response

We thank the referees for their constructive criticisms and the many suggestions to im-
prove our manuscript. In the revised version we followed the referees’ suggestions and we
modified or rewrote parts of the manuscript. The changes are summarized in the "List of Changes" section. We are confident we touched upon all requested points and we remit to the referees for their judgement on the revised paper.

---

## Round 1 · List of Changes

• MAJOR CHANGES: We expanded the introduction, adding more references to brick
wall quantum circuits in different forms, clarifying notation issues and refining some
parts to improve readability. In particular, we strengthened the motivation which
brought us to investigate this class of quantum circuits, highlighting the underlying
free fermionic structure and the global fermionic symmetry. We clearly formulated
challenges and open questions for this class of circuits. We also pointed out the con-
nection to QCAs, which corroborates our motivation to study regularized versions
of QFTs endowed with supersymmetry. Furthermore, we followed the suggestions of
the referees and we completely restructured section 5. We ran TEBD simulations of
quench dynamics for large sizes and many layers of our brick wall unitary, focussing
on the time evolution of single site magnetizations. We compared the findings with
analytical reasoning based on the spectral structure that we uncovered, and showed
that the equilibration of magnetizations can be captured by a GGE based on the
free fermionic structure. We report a remarkable connection of our work to the
recent work of Zadnik et al. We restructured the outlook in accordance with the
changes made and added an appendix with some details of the quench dynamics
for PBC.
• MINOR CHANGES: We improved overall readability (especially in parts pointed out
by the referees), corrected typos, expanded comments on graded tensor products
and their relation to JW-transformations, expanded the explanation on the topo-
logical insulator part, explained further the role of global fermionic symmetry in the
criticality of the Hamiltonian limits, clarified the difference between Hamiltonian
limit and UF, explained unclear notations.

---

## Round 2 · Author Response

List of changes
- clarified how the commutation relation works for the boundary matrix K -made spin notation consistent in eq. 1.2 -corrected a typo in eq. 2.4 to clarify how parity affects local operators in graded tensor products -enlarged fontsize for numerical plots

---

## Round 2 · List of Changes

- clarified how the commutation relation works for the boundary matrix K -made spin notation consistent in eq. 1.2 -corrected a typo in eq. 2.4 to clarify how parity affects local operators in graded tensor products -enlarged fontsize for numerical plots

---

## Editorial Decision

published